# Synthetic analysis of chromatin tracing and live-cell imaging indicates pervasive spatial coupling between genes

Christopher H Bohrer, Daniel R Larson*

Laboratory of Receptor Biology and Gene Expression, Center for Cancer Research, National Cancer Institute, National Institutes of Health, Bethesda, United States

**Abstract** The role of the spatial organization of chromosomes in directing transcription remains an outstanding question in gene regulation. Here, we analyze two recent single-cell imaging methodologies applied across hundreds of genes to systematically analyze the contribution of chromosome conformation to transcriptional regulation. Those methodologies are (1) single-cell chromatin tracing with super-resolution imaging in fixed cells; and (2) high-throughput labeling and imaging of nascent RNA in living cells. Specifically, we determine the contribution of physical distance to the coordination of transcriptional bursts. We find that individual genes adopt a constrained conformation and reposition toward the centroid of the surrounding chromatin upon activation. Leveraging the variability in distance inherent in single-cell imaging, we show that physical distance – but not genomic distance – between genes on individual chromosomes is the major factor driving co-bursting. By combining this analysis with live-cell imaging, we arrive at a corrected transcriptional correlation of $\phi \approx 0.3$ for genes separated by < 400 nm. We propose that this surprisingly large correlation represents a physical property of human chromosomes and establishes a benchmark for future experimental studies.

*For correspondence:
dan.larson@nih.gov

Competing interest: The authors declare that no competing interests exist.

## Editor's evaluation

In this article, Bohrer and Larson revisit previously published imaging datasets in order to tackle a long-standing question in modern genome biology: does the physical proximity of transcribed genes correlate with their co-expression? The authors provide convincing evidence to deduce that when a pair of loci are brought within sufficiently low physical 3D proximity (unrelated to their genomic distance) they are more likely than expected to be co-expressed. This is a result of potentially fundamental importance.

## Introduction

The role of spatial heterogeneity in the nucleus in relationship to gene regulation is an enduring question in cell biology (**Bohrer and Larson, 2021**). Heterogeneity or compartmentalization is visible at all length and genomic scales, starting from gene loops and proceeding through enhancer–promoter interactions, topologically associated domains, A/B compartments, chromosome territories, up to inter-chromosomal interactions such as the nucleolus, Cajal bodies, and histone locus bodies, and extending to prominent nucleus-wide features such as lamin-associated domains and heterochromatin (**Misteli, 2020**). The synergy between microscopy (mostly light microscopy but also electron microscopy; **Ou et al., 2017**) and chromosome conformation capture approaches has led to fundamental insights of how molecular features drive genome organization, the influence they have on gene regulation, and the extent to which genome organization varies within individual cells.

Yet, the chromatin–transcription relationship at length scales smaller than the wavelength of visible light (~500 nm) remains challenging to dissect. Foundational work from Cook and colleagues introduced the notion of the transcriptional factory. Transcription factories are areas with an enrichment of transcription machinery where genes are thought to be transiently bridged to enable efficient transcription (*Feuerborn and Cook, 2015*). Ensemble chromosome conformation capture seems to support this model by revealing that promoter–promoter contacts (smaller than 1 Mb) form as transcription levels increase (*Hsieh et al., 2021*; *Zhu and Suh, 2020*; *Hsieh et al., 2020*; *Levo et al., 2022*). The model is that actively transcribed genes are positioned to transcription factories. The prediction is that genes that are close in 3D space (nm) will 'feel' the same enrichment in transcription machinery and exhibit correlated transcriptional bursts. Indeed, genes on the same chromosome (*Deng et al., 2014*; *Sun and Zhang, 2019*; *Tian et al., 2020*; *Quintero-Cadena and Sternberg, 2016*; *Xu et al., 2019*) and genes that share the same (ensemble) topologically associated domain (*Tarbier et al., 2020*) are more co-expressed in individual cells (RNA). However, correlations were not seen between nascent transcripts (*Levesque and Raj, 2013*) and the genomic distance between genes was found to show a more dominant role in RNA co-expression than Hi-C contact frequency (*Sun and Zhang, 2019*). Furthermore, single-cell RNA-seq showed little to no difference in correlation between genes from the same chromosome with an increased contact frequency, given a similar genomic distance between the two, bringing the strength of the hypothesis into question (*Tarbier et al., 2020*).

This static factory view was supplanted by one in which local heterogeneity of the transcription machinery was due to dynamic assembly and disassembly (*Cisse et al., 2013*; *Cho et al., 2018*; *Henninger et al., 2021*). Thus, the 'factory' was not a fixed assemblage but rather a transient and movable conglomeration of RNA polymerase II, general transcription factors, and nascent RNA that arose in connection to active transcription units. It is clear that these diffraction-limited spots observed in the fluorescence microscope exchange constituents with the surrounding nucleoplasm. However, the number of terms used to describe these spots – 'factories,' 'foci,' 'hubs,' 'clusters,' 'speckles,' 'compartments,' 'condensates,' 'phases' – emphasizes the lack of a consensus model in the field. Further, it should be noted that many of the utilized super-resolution methodologies are prone to artifacts (*Bohrer et al., 2021*). Consequently, the physical interactions between protein, DNA, and RNA and the dynamic changes in chromosome structure that precede RNA synthesis are hotly debated.

Recent advances in single-cell imaging shed light on these questions and motivate the fully theoretical analysis in this paper. First, the development of chromatin tracing of an entire chromosome using super-resolution light microscopy provides a spatial map of the chromatin fiber at ≈100 nm resolution (*Su et al., 2020*; *Hu and Wang, 2021*). When coupled with single-molecule fluorescence in situ hybridization (smFISH) to look at nascent RNA, one can then connect chromatin conformation to transcriptional activity with single-cell resolution (*Su et al., 2020*). Specifically, the nascent transcription state of ~80 genes as well as the 3D centroid positions of 651 50 kb chromosomal segments was quantified for thousands of individual chromosomes in IMR90 cells (*Figure 1A*). Second, the application of single-cell imaging of nascent RNA in living cells provides critical information on temporal heterogeneity to interpret the observations of spatial heterogeneity. For example, transcriptional bursting of human genes expressed in their native genomic context can be monitored with high spatial and temporal precision for hours (*Rodriguez et al., 2019*; *Wan et al., 2021*).

Here, we take advantage of two single-cell datasets – chromatin tracing in fixed cells and nascent RNA imaging in living cells to address two questions: (1) Do genes reposition upon transcriptional activation? (2) Do genes in spatial proximity show correlations in transcriptional activity? Our analysis indicates that with transcription, chromatin adopts a constrained structure and the gene is positioned toward the centroid of the surrounding chromatin. We then probed the distances between genes and found that genes are positioned closer to each other with transcriptional bursts when the genomic distance between them below 5 Mb, and genes were positioned farther away from each other with transcription if the genomic distance was above 5 Mb. Importantly, by capitalizing upon the fluctuations of distances between genes on individual chromosomes, we found that the physical distance between genes on individual chromosomes is the major factor driving the transcriptional co-bursting between genes. By incorporating temporal information from live-cell imaging of active genes (duration of active periods and mobility of active genes), we can infer the correlation between transcriptional bursts for proximal genes to be $\phi \approx 0.3$. Overall, our synthetic analysis of these two single-cell

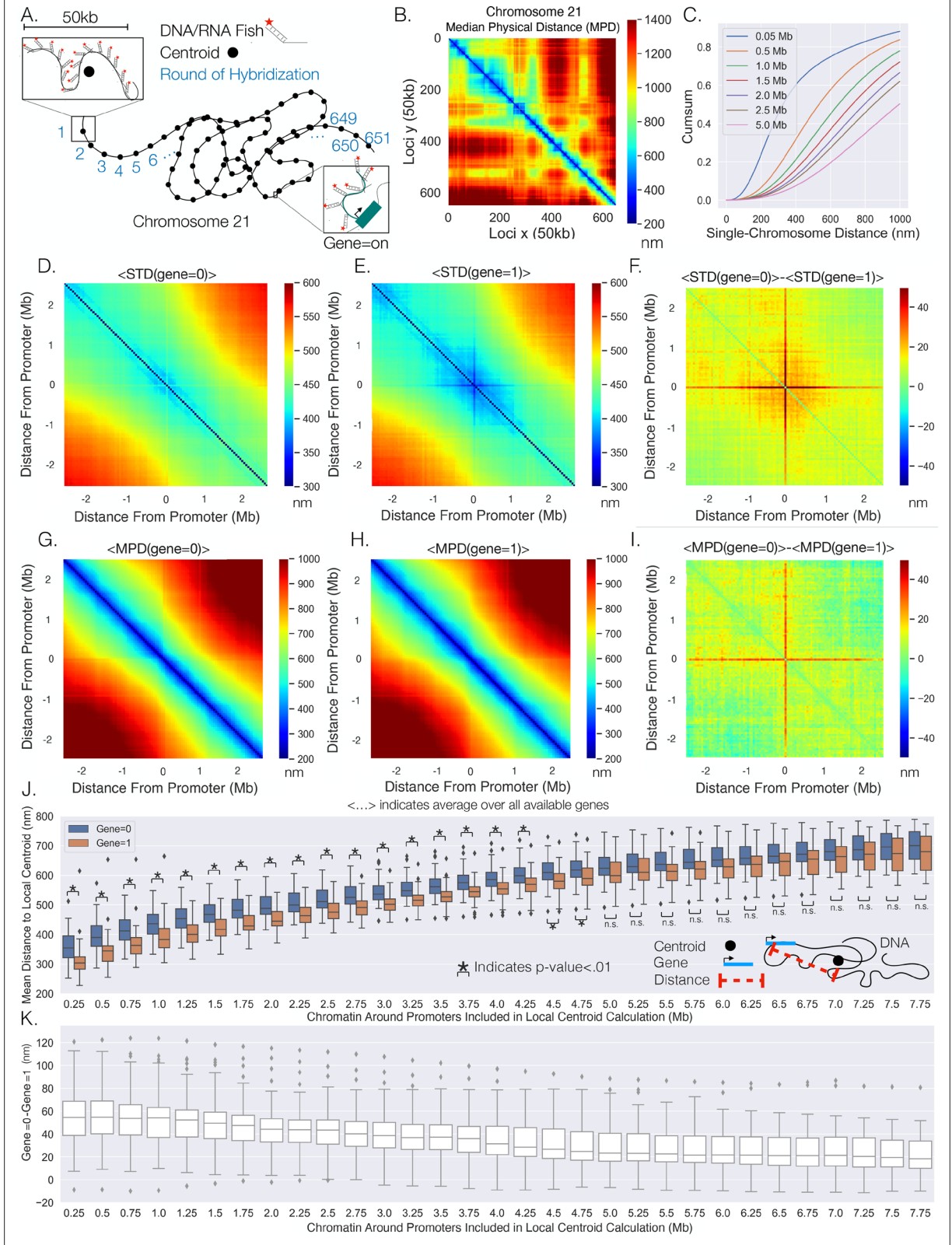

**Figure 1.** Transcription confines chromatin and active promoters are located toward the centroid of their surrounding chromatin.
(**A**) An illustration of the chromatin-tracing data where each chromosomal locus is imaged through different rounds of hybridizagtion and the centroid of each 50 kb region is determined. Nascent RNA FISH was used to classify genes into 'on' (1) or 'off' (0) according to their transcriptional state. (**B**) The median physical distances (MPD) between all loci determined on chromosome 21. (**C**) The cumulative distribution function of the distance between

*Figure 1 continued on next page*

*Figure 1 continued*

chromosomal loci separated by various genomic distances – all loci with a given genomic distance were used to generate these distributions. (**D**) An aggregate analysis, calculating the standard deviation (STD) of the distances between chromosomal loci for chromosomes where a gene = 0, centered around the loci containing the promoter, and then averaging over all genes. (**E**) The same as (**D**) but with gene = 1. (**F**) The difference in the average centered STD in (**D**) and (**E**). (**G**) Similar to (**D**) but quantifying the MPD instead of the STD. (**H**) The same as (**G**) but for chromosomes where gene = 1. (**I**) The difference between the average centered MPD in (**G**) and (**H**). (**J**) The mean distances between chromosomal loci containing genes to the centroid of the surrounding chromatin when the genes were either on (1) or off (0) vs. the amount of chromatin around the promoter included in the centroid calculation. There is also an illustration of this calculation in the far-right corner to aid interpretation. (**K**) The difference between the mean distances to the local centroid when gene = 0 and gene = 1, showing the results in (**J**) on a gene-by-gene basis. Boxplots show quartiles and whiskers expand to 1.5× interquartile range, black diamonds are outliers. Significance was defined as a p-value <0.01 with a t-test (Appendix 1). The analysis was done on ≈7600 individual chromosomes and 80 different genes.

datasets indicates that indeed genes do reposition upon activation and show concomitant correlation between individual transcriptional bursts.

## Results

## Active promoters are positioned to locations defined by chromatin organization

To investigate spatial changes in the chromatin fiber for active and inactive genes, we reanalyzed data from combined super-resolution imaging of DNA and RNA FISH (*Su et al., 2020*). We performed a spatial metagene analysis consisting of 'centering' the chromatin around the promoter of the each gene, quantifying the standard deviations (STD) of the distances between the chromosomal loci, and then averaging over all available genes. Note, we utilized the centroid position of the chromosomal segment that contained the transcriptional start site of each gene as the location of the promoter for the gene and only utilized the chromosome tracing by sequential hybridization data (*Su et al., 2020*). This analysis was done for chromosomal segments where genes were 'off' (0) or 'on' (1) (*Figure 1D and E*) – we utilize Boolean logic (0 or 1) throughout to describe transcription states based on the absence (0) or presence (1) of nascent RNA. We observed that chromatin centered around the promoter shows less variability while transcribed, again as determined by the presence of nascent RNA. To more clearly visualize distinctions between chromatin configuration ± nascent RNA, we quantified the difference and found that the distances from a promoter to the surrounding chromatin are more restricted with transcription, indicated by a cross-shape pattern on the heatmap (*Figure 1F*).

The change in confinement could be the result of repositioning active genes to a different nuclear environment. To probe whether gene positioning varies with transcription, we performed a similar analysis but quantified the median physical distance (MPD) between chromosomal loci with and without transcription and quantified the average over all available genes (*Figure 1G and H*). Again, we quantified the difference between them and found a similar red cross (*Figure 1I*), suggesting that when a gene is active the promoter is on average closer to the surrounding chromatin and the distances between nonpromoter chromosomal segments are unperturbed.

It is conceivable that repositioning is due to enhancer–promoter proximity that might precede transcription activation: the smaller average MPD to the surrounding chromatin with transcription could be due to genes only being active when near surrounding specific enhancers. To investigate, we used the density of H3K27Ac as a proxy for enhancer activity. We quantified the density of H3K27Ac ChIP-seq reads within each 50 kb segment for IMR90 cells using previously acquired data (Appendix 1; *ENCODE Project Consortium, 2012*). This analysis resulted in varying densities of H3K27ac throughout Chr21 and is shown in *Appendix 1—figure 1A*. We then partitioned the H3k27ac density into four groups (low, med, high, very high) and investigated the average MPD of each gene to all other loci with and without transcription. Like before (*Figure 1*), we observed that a gene was indeed closer to the other individual loci when transcriptionally active, but the MPD change did not show a general difference with H3K27ac enrichment when compared to other loci lacking H3K27ac (*Appendix 1—figure 1B*), suggesting that the observed repositioning may not be a result of enhancer–promoter interaction.

Intuitively, a possible reason for the distance to decrease to surrounding chromatin with transcription (on average) is if a gene is located closer to the centroid of the surrounding chromatin for single chromosomes when active. To test this supposition, we calculated the mean distance of the

promoter of the gene to the centroid of the surrounding chromatin with and without transcription (*Figure 1J*). The centroid was calculated for windows of various genomic size around each gene – that is, for a 0.5 Mb chromatin region, 0.25 Mb on both sides of the gene promoter were included in the centroid calculation. Tellingly, we found a definitive difference between active promoters (1) and inactive promoters (0): the active promoters were closer to the centroids of the surrounding chromatin (*Figure 1J*). Note that the mean distance from a local centroid to an inactive promoter gives one an idea to natural spread of the chromatin. To understand this phenomenon on a gene-by-gene basis, we quantified the difference between the active promoter and inactive promoter for each gene (*Figure 1K*). We found that even though there are overlaps in the distributions in *Figure 1J*, nearly every gene was closer to the centroid with nascent transcription, suggesting a general phenomenon. Overall, these results indicate that transcriptionally active genes are located toward the centroid of surrounding chromatin.

We then sought to assess whether the positioning of the genes toward the centroid was dependent upon transcriptional activity. To investigate, we partitioned the available genes into low activity or high activity depending upon whether fractional occupancy was below or above the median, and then performed the above analysis on each subset of genes. That is, the activity of a gene was determined from the fraction of chromosomes where that gene was active. Interestingly, we found that

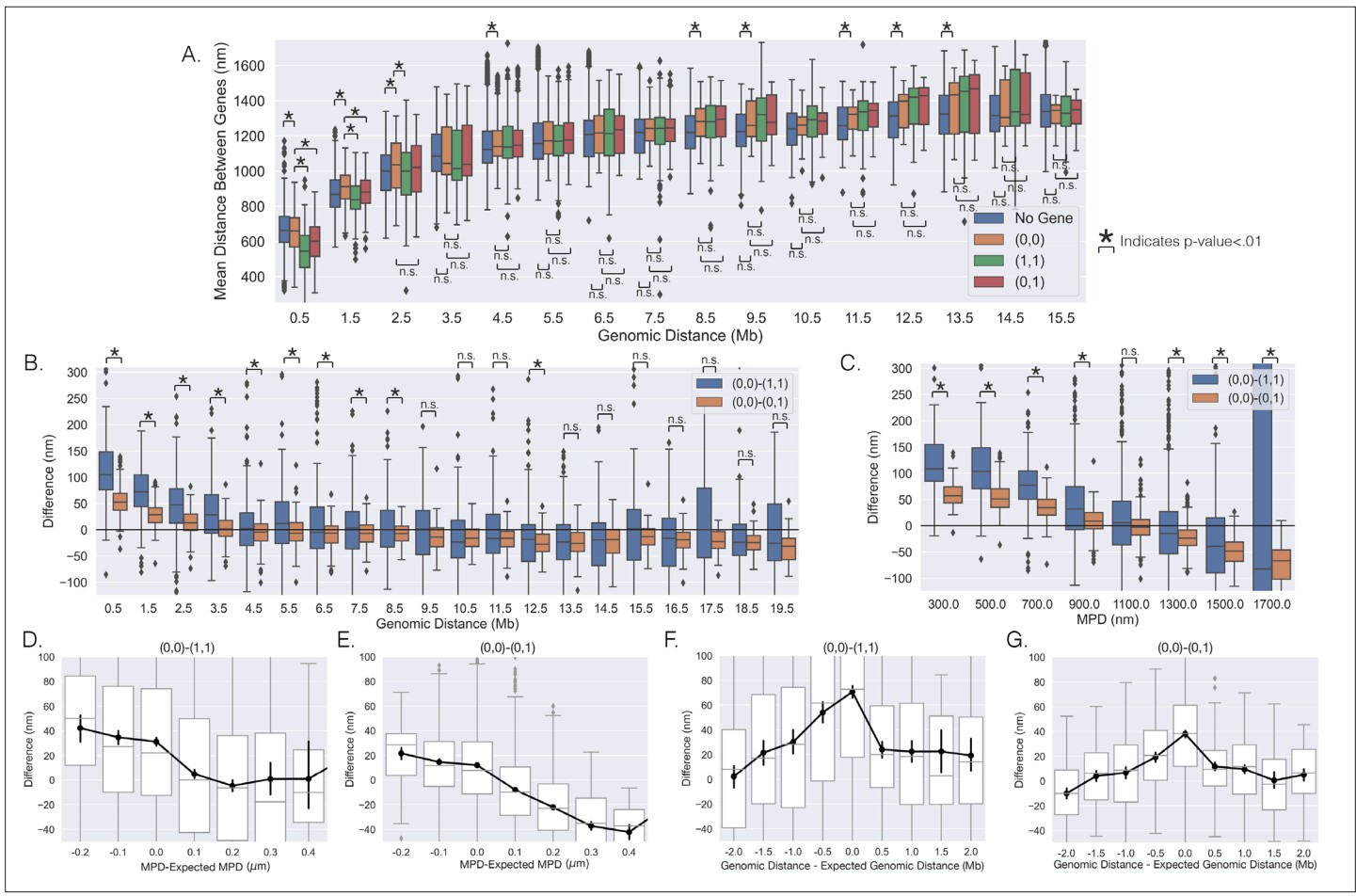

**Figure 2.** The distances between genes vary with transcription on individual chromosomes. (**A**) The mean distances between genes vs. the genomic distance for when both genes were (0,0),(1,1), (0,1), and the mean distances between loci not containing the investigated genes. Boxplots show quartiles and whiskers expand to 1.5× interquartile range, black diamonds are outliers. (**B**) The difference between the scenarios shown in (**A**), showing the difference in mean distance on a gene pair by gene pair basis, and a black line is shown to aid in visualization of zero. (**C**) The same analysis as in (**B**) but vs. the median physical distance (MPD) between the genes. (**D–G**) The difference shown in (**B**) and (**C**) but vs. either the MPD minus the expected MPD or the genomic distance minus the expected genomic distance (see text). Boxplots show quartiles and whiskers expand to 1.5× interquartile range, black diamonds are outliers. Black lines and dots are means and error bars are SEM from bootstrapping (Appendix 1). Significance was defined as a p-value < 0.01 with a *t*-test (Appendix 1).

high-activity genes were both less variable (*Appendix 1—figure 2A*) and showed greater movement with active transcription when compared to the low-activity genes (*Appendix 1—figures 2B and 3*). Upon closer inspection (*Appendix 1—figure 3A*), the greater movement for the high-activity genes was not so much due to a different distance to the local chromatin centroid when active but was instead due to larger distances from the centroid when inactive – this is illustrated by the first genomic distance bin in *Appendix 1—figure 3A* by comparing the first genomic distance bin of the low-activity genes to the high-activity genes. In brief, these results suggest that these processes additionally vary depending upon a genes activity level.

Having considered genes individually based on activity (first order moments), we next sought to quantify higher-order moments such as pairwise interactions in promoter–promoter distances based on transcriptional activity. We first quantified the average distances between promoters when [both genes were off, (0,0)], [both were on, (1,1)], [one was off and one was on, (0,1)], and quantified them as a function of the genomic separation between them (*Figure 2A*). We also quantified the average distances between chromosomal loci that did not contain the investigated genes as a reference control (*Figure 2A*). We found that the distances between genes were consistently smaller with transcription for short genomic distances (<1.5 Mb), as evidenced by the significant decrease in the (0,1) and (1,1) interactions compared to the (0,0) interaction. When we compared (0,0) to the no gene control, we saw essentially no difference. We note that the means of the samples were statistically different in some cases (i.e., no gene to (0,0)), potentially indicating that the distances between the genes are potentially different even when inactive (*Figure 2A*). Still, overall, these results suggest transcriptional bursting (or a consequence of bursting) is correlated with the formation of promoter–promoter contacts.

To probe the distance changes on a gene pair by gene pair basis, we first calculated the mean distance between inactive genes on the same chromosome (0,0) and then subtracted the mean distance between the genes when active ((1,1) or (0,1)) – similar to the analysis in *Figure 1K*. This analysis is shown as a function of the genomic distance between genes in *Figure 2B*. For genomically proximal genes, we observed that when both genes were active the mean distances between the promoters were indeed closer to each other. When we compared the (0,0)–(0,1) to (0,0)–(1,1), the later difference was approximately twice the former difference. Interestingly, we observed that as the genomic distance increased, the difference for both seemed to approach a negative value, suggesting that sufficiently separated genes are positioned to different locations with transcription. However, the spread within the boxplots suggests much variability in whether genes are positioned toward the same or different location with transcription. Overall, these analyses provide strong evidence that the spatial separation between genes depends on individual transcriptional bursts.

These analyses suggest a characteristic genomic length scale over which pairwise interactions might occur. However, since genomic distance and physical distance between chromosomal segments are obviously correlated (*Sun and Zhang, 2019*; *Bintu et al., 2018*; *Su et al., 2020*), either might define the length scale and drive repositioning with transcriptional bursting. To probe the general impact of MPD, we characterized the positioning of genes toward the same or different location with transcription based on the 3D distance between the genes. Note that this analysis is only possible with microscopy datasets such as this one (*Su et al., 2020*). We performed the previous analysis as a function of the MPD between the genes (*Figure 2C*) and found a strong decay with increasing MPD. The (0,0)–(0,1) resulted in a strong majority of values being negative for MPD above 1300 nm, indicating that the genes move away from each other with bursting above this spatial threshold. The (0,0)–(1,1) had a majority of negative values for MPD above 1300 nm but the proportion with positive values was higher.

Probing further, to disentangle the dependence of this movement on genomic distance and/or MPD, we quantified how deviations from the expected influenced repositioning. Given the stronger trend with the MPD, we first quantified the difference as a function of the MPD minus the expected MPD. The expected MPD was calculated utilizing all chromosomal loci and was defined as the average MPD for each genomic distance ('Methods'). We found that for both scenarios a smaller than expected MPD resulted in genes moving toward each other with transcription and a larger than expected MPD led to the genes moving away from each other (*Figure 2D and E*), though the latter was less clear for the (0,0)–(1,1). These results suggest that the positioning of genes in physical space influences the outcome of pairwise interactions: genes which are close to each other (MPD <1100 nm) move

closer when bursting, and genes that are far from each other separate when bursting. Similarly, to investigate whether the genomic distance plays a role, we performed the analysis but as a function of the genomic distance minus the expected genomic distance — the genomic distance given the MPD ('Methods'). We found that the analysis did not have a monotonic trend and instead peaked at zero (*Figure 2F and G*). If there were a simple relationship between genomic distance and repositioning, one would expect a monotonic trend and therefore it seems unlikely that genomic distance drives this phenomenon. Additionally, we found that the zero peak was enriched for gene pairs with low MPDs – as we just demonstrated: low MPDs lead to genes moving toward each other (*Figure 2D and E*). In summary, these results suggest that the MPD is predictive of whether genes move toward or away from each other with transcription.

Lastly, we sought to probe the extent to which this phenomenon was dependent upon transcriptional activity (low vs. high as described above). As before, we performed the same analysis but on the two groups of genes separately. Again, the distance change between genes was stronger for more active genes, suggesting these processes also vary depending upon the transcription activity level (*Appendix 1—figure 4*). Of note for high-activity genes, nearly all of them move away from each other when they were separated by large MPD (>1300 nm), suggesting the process of moving to a different location for transcription may be more deterministic for highly active genes (*Appendix 1—figure 4E*).

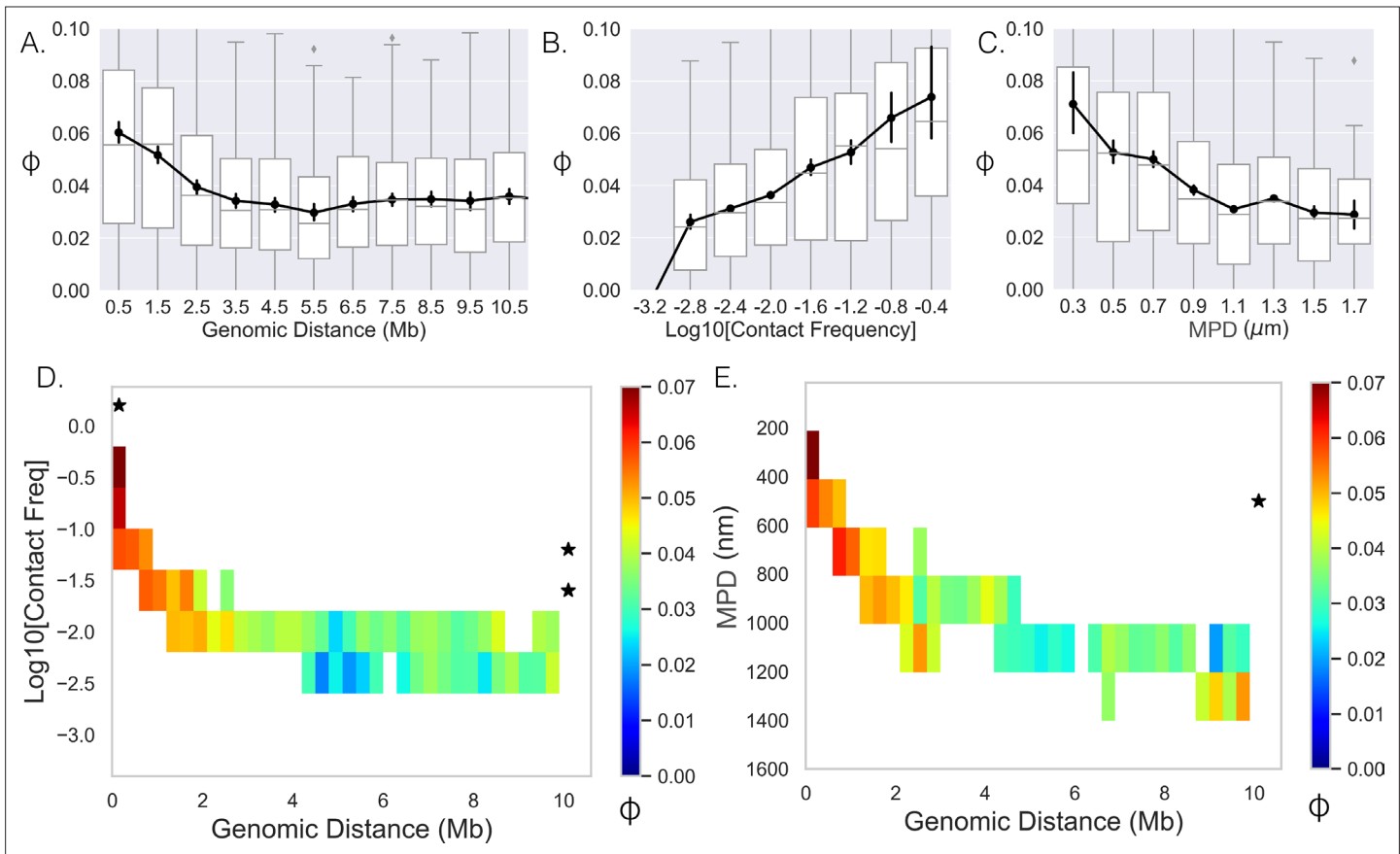

**Figure 3.** Limited variability prevents quantification. (**A–,C**) The Spearman correlation coefficient between genes as a function of genomic distance, contact frequency, and median distance. Black lines and dots are means and error bars are SEM from bootstrapping (Appendix 1), boxplots show the quartiles as above. (**D**) Average correlation coefficients of genes given that their genomic distance and contact frequencies were within a specific range. (**E**) Average correlation coefficient of genes given that their genomic distance and median distance were within the specific range. An * illustrates whether the average correlation coefficients along that dimension are correlated (p-value<0.01) (Appendix 1).

## Physical distance – but not genomic distance – correlates with co-expression

Our analysis of the DNA/RNA FISH dataset indicates that spatial gene positioning is correlated with transcriptional activity both in isolation (repositioning of individual genes with transcription) and in pairwise interactions. One can conceptualize the conclusions of this analysis as understanding spatial position given the transcriptional state. In other words, knowledge of transcription state imparts knowledge of spatial position. We next turned to the inverse question of whether correlations exist between nascent RNA (nRNA, transcriptional state) based on spatial proximity. To do so, we quantified the $\phi$ correlation coefficient ('Methods') between genes on individual chromosomes (*Figure 1A*) and plotted it as a function of the genomic distance (*Figure 3A*). Note, due to the binary nature of the data (0 or 1), the $\phi$ correlation coefficient is equivalent to the Pearson and Spearman. With approximately a twofold increase at smaller genomic distances, the correlation showed a monotonic decay with increasing genomic distance – the 0.025 plateau persisted with even higher genomic distances (data not shown). The increase in co-expression above the asymptotic baseline persists to ≈2 Mb. To determine whether ensemble-chromatin structure is what dictates co-expression, we further quantified the correlation as a function of the contact frequency (*Figure 3B*) and the MPD between their chromosomal segments (*Figure 3C*). Here, we defined the contact frequency between two genes as the proportion of chromosomes with distances less than 200 nm between the genes' chromosomal segments using the chromatin-tracing data. We observed the predicted monotonic behavior with the average correlation reaching a minimum around 0.025.

We then attempted to separate the effects of contact frequency/MPD from genomic distance on the observed correlation, and proceeded to hold one variable constant and quantify the correlation as a function of the other. To do this, we calculated the mean correlation given that the contact frequency/MPD and genomic distance between the genes were within a specified range (*Figure 3D and E*). Note that we only included averages if more than 40 data points could be used to calculate the mean. The two showed similar behavior and both had a narrow range for specific genomic distances, making it difficult to uncouple the variables of contact frequency and mean physical distance. For example, we only observed an MPD of 200–400 nm for genomic distances much less than 1 Mb; therefore, we could not determine how the correlation varies with increasing genomic distance for these values. Moreover, most columns and rows did not show significant p-values. In summary, while there is correlation at the nascent RNA level, the limited variability in ensemble-chromatin structure for specific genomic distances obscured the relative contributions of genomic distance, contact frequency, or MPD to co-expression.

A primary advantage of the single-cell dataset (*Su et al., 2020*) is the ability to leverage the large fluctuations of distances between loci across the population (N ≈ 7600 chromosomes) (*Figure 1C*). We first quantified the correlation between nascent RNA for genes given that their physical distances were within a specific range, which showed a similar monotonic behavior (*Figure 4A*). When calculating these correlation coefficients, we only included gene pairs for specific single-chromosome distance ranges when there were at least 100 chromosomes where the distance between the genes was within that range. We then quantified the mean correlation given that their single-chromosome distance and genomic distance were within specified ranges (*Figure 4B*). Again, we only included averages if more than 40 data points (gene pairs) could be used to calculate the mean. Notably, we observed that co-expression of genes was correlated with the single-chromosome distance between those genes (columns, *Figure 4B*). In contrast, we observed no correlation between co-expression and genomic distance (rows). There appeared to be a general decay for the columns with increasing single-chromosome distance, more closely resembling the curve in *Figure 4A*, while the rows did not show the behavior. These observations are further solidified by calculations of statistical significance (*Figure 4B*).

In summary, these results indicate that co-expression – as quantified through correlations in nascent RNA – is driven by the physical distance between genes on individual chromosomes, uncoupled from genomic distance, which shows no statistical correlation with co-expression.

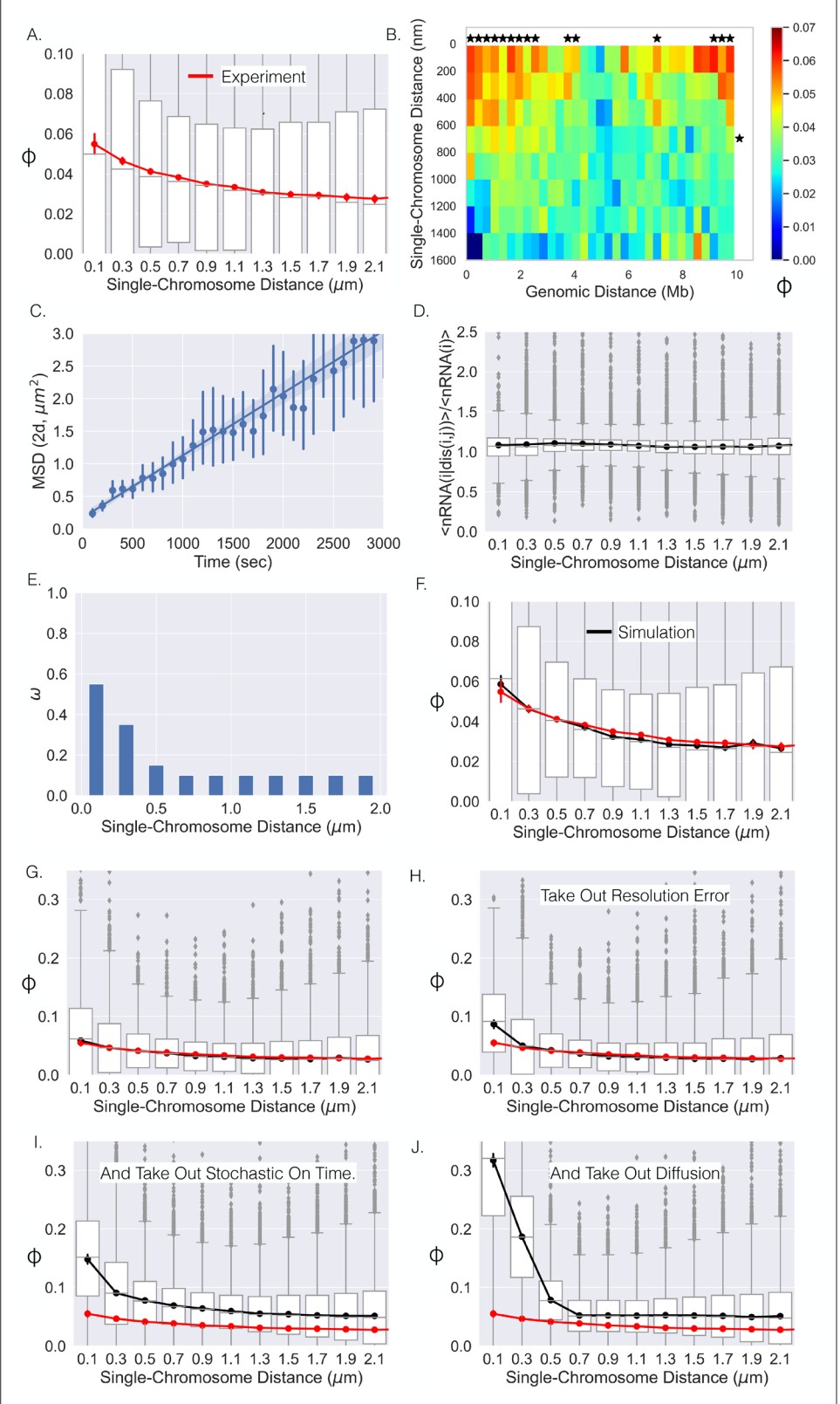

**Figure 4.** Single-chromosome distance dictates nRNA correlation. (**A**) The correlation coefficients between genes as a function of single-chromosome distance. (**B**) Average correlation coefficients of genes given that their genomic distance and single-chromosomal distance were within a specific range. An * illustrates whether the average correlation coefficients along that dimension are correlated (p-value<0.01) (Appendix 1). (**C**) The mean-

*Figure 4 continued on next page*

*Figure 4 continued*

squared displacement of active *TFF1*, the fitted line, and 95% CI shaded (error bars are individual 95% CIs). (**D**) The average number of chromosomes with nRNA for gene *i* given the distance between gene *j* and *i* divided by the average with all distances. (**E**) The optimal $\omega$ function for the model that results in the black curve in (**F**). (**F**) The correlation–distance relationship for all pairs of genes from the simulation utilizing the $\omega$ function in (**E**). The boxplots here are from simulation, red curve is shown for reference and is the experimental data from (**A**). (**G**) The same as (**F**) but on a different scale. (**H**) The results of the simulation without resolution error of the experiment. (**I**) Simulation results without resolution error and with nRNAs having a deterministic on time. (**J**) Simulation results without resolution error, with deterministic on times, and no chromatin diffusion for all pairs of genes.

## Chromosome dynamics can obscure the true correlation between physical proximity and gene co-expression

The single-cell DNA/RNA FISH approach provides exceptional spatial resolution coupled with transcriptional activity, but a potential issue with fixed-cell methodologies is the lack of temporal information. For example, in terms of quantifying the distance dependence on co-expression, the lack of time-resolved locus position data could distort the observed distance co-expression relationship. First, the motion of the genes within the on time (defined here as the time it takes for the nascent RNA to dissociate from the DNA) obscures the measurement of the distance at the beginning of a transcriptional co-burst. Second, the stochasticity of the on time would similarly lead to a decrease in the observed co-expression – that is, even if two genes burst at the exact same time, the nascent RNA from one gene will dissociate before the nascent RNA of the other gene, leading to the detection of one and not the other, again decreasing the correlation measured in fixed cells. Third, the error due to the localization precision of the experiment would also distort the distance co-expression curve due to the error in knowing the true distance. Overall, these three sources of noise have the potential to change both the amplitude and distance dependent decay of the co-expression correlation coefficient. Therefore, we utilized a theoretical approach to infer the instantaneous distance co-expression relationship analogous to that shown in *Figure 4A* and to thereby understand the contribution of dynamic and temporal fluctuations in gene position and activity. The approach is based on coupling measurements of locus diffusion and activity generated from live-cell imaging of nascent RNA with the fixed cell measurements analyzed thus far. Here, we first discuss our theoretical approach and then our results.

We sought to link the information from live-cell experiments with that of fixed-cell experiments by incorporating the motion of chromatin into our model. Chromatin has been suggested to show confined diffusion (*Marshall et al., 1997*; *Chubb et al., 2002*; *Chen et al., 2013*; *Bronshtein et al., 2015*), but this phenomenon is generally quantified over relatively short timescales of <10 min. Considering the on time of a human gene – as measured by the dwell time of nascent RNA – is approximately 10–15 min (*Wan et al., 2021*), we sought to monitor the diffusion of an active gene over a longer timescale. We first utilized the live-cell transcriptional bursting data of *TFF1* from *Rodriguez et al., 2019*. This data consists of the spatial coordinates of multiple bursting *TFF1* alleles through time in individual cells, allowing us to quantify the motion of one allele relative to the other (*Chubb et al., 2002*). Importantly, time-lapse imaging of multiple alleles naturally corrects for cell movement over these long timescales. We quantified the mean squared displacement (MSD, 'Methods') over a timescale of 3000 s and found that the MSD could be fit with a straight line (*Figure 4C*), suggesting Brownian motion of active genes over these timescales (*Bohrer and Xiao, 2020*). We computed a diffusion coefficient of $D_{TFF1} = .25 \times 10^{-3} \mu m^2/s$, which is comparable to previous results (*Chubb et al., 2002*). We subsequently performed a similar analysis with the previously published live-cell transcriptional bursting data of four different genes and obtained similar results but with slightly varying diffusion coefficients (*Appendix 1—figure 5*; *Wan et al., 2021*). Taking into account the multiple diffusing alleles within the *TFF1* data (Appendix 1), the four diffusion coefficients of the single-locus genes range from about $.25 \times D_{TFF1}$ up to $1 \times D_{TFF1}$. Lastly, we ultimately decided to proceed with the diffusion coefficient of *TFF1* due to the natural cell movement correction and the relative similarity with the other diffusion coefficients.

We chose to utilize the over-dampened Langevin equation to model the temporal dynamics of the distance between genes located on the same polymer. The model describes the time-dependent distance between loci using an arbitrary energy potential of interaction (see 'Methods') – without

the effect of the potential the model exhibits Brownian motion with the determined diffusion coefficient. For each gene pair, we empirically determined a potential that 'biases' the distance motion so the steady-state distribution matches the empirically determined distance distribution ('Methods'). We did this using the equivalent Fokker–Planck equation, which allowed us to directly convert the empirically defined distance distributions into the potential ('Methods'). The central advantage of this approach is that it accounts for the unique distance distributions between the various gene pairs on the same chromosome, the diversity of which can be clearly seen with the MPDs in *Figure 1B*. The diverse distance distributions result from a multitude of complex context-specific forces that are not considered in the classical polymer models (*Osmanović and Rabin, 2017*; *Vivante et al., 2020*). Even with the inclusion of additional factors in polymer models (exp. loop extrusion), reproducing accurate distance distributions is difficult (*Gabriele et al., 2022*) – and would be even more difficult here due to lack of knowledge as to the underlying forces. Also, more simple first-order approximations of the Langevin equation have been utilized to model the viscoelastic properties of chromatin (*Vivante et al., 2020*), which has been shown to adequately determine the potential of the Rouse chain (*Amitai et al., 2015*). Again, we emphasize that these gene-specific terms were determined empirically ('Methods').

The stochastic dwell time of nascent RNA is due to variability in the processes of elongation, termination, and splicing. We incorporate this variability in our analysis by setting the nascent RNA decay probability per second (propensity) equal for all genes ($P_d$) with a characteristic on time equal to $\approx$13 min. This assumption is motivated by our recent work on high-throughput imaging of hundreds of human genes labeled at their endogenous loci using MS2 stem loops – where it was found the majority of genes had an average on times between 10 and 15 min (*Wan et al., 2021*). Again, we note that this is an assumption due to our lack of temporal information.

Next we introduce a phenomenological model intended to capture the empirical features of co-expression as observed in the fixed cell datasets. First, we quantified the average fraction of chromosomes with nascent RNA present for gene $i$ as a function of the distance between each pair of genes (genes $i$ and $j$), normalized by the average fraction of chromosomes with nascent RNA present for gene $i$ over all distances. This metric is a proxy for the burst frequency and was calculated for each gene for all possible gene pairs. The reasoning is that if this metric is higher at smaller distances, it would suggest that the bursting frequency is dependent upon the distance between genes, hence leading to the higher correlation values at smaller distances. Surprisingly, we found that on a distance binning scale of 200 nm, the metric did not vary, suggesting that the bursting frequency does not generally change as function of distance between genes at this scale (*Figure 4D*). Therefore, we set the probability of nascent RNA production per second equal to a constant for each gene ($i$), $P_i^{tot}$, which we determined empirically for each gene ('Methods'). To account for co-expression, we modeled nascent RNA production as coming either from a co-burst or from an individual burst, where the likelihood that a co-burst or an individual burst occurs is dependent upon the distance between the two genes ('Methods'). More specifically, the fact that a pair of genes have differing expression levels allowed us to model the proportion of transcription events that are co-bursts with the incorporation of the function $\omega(r_{ij}(t))$, which is a function of distance between the genes and ranges between 0 and 1. For a pair of genes where the burst frequency of gene $i$ is less than gene $j$, $\omega(r_{ij}(t))$ is the proportion of gene i's transcriptional bursts that are co-bursts at each distance ('Methods'). If the expression levels of the two genes are approximately equal, $\omega(r_{ij}(t))$ is equal to the proportion of bursts that are co-bursts at a given distance for both genes.

Overall, with a single coupling function ($\omega(r_{ij}(t))$), we modeled all pairs of genes with the following stochastic reactions utilizing the Gillespie algorithm (*Gillespie, 1977*):

$$0 \xrightarrow{P_{ij}(r_{ij}(t))} nRNA_i + nRNA_j,$$

$$0 \xrightarrow{P_i(r_{ij}(t))} nRNA_i,$$

$$0 \xrightarrow{P_j(r_{ij}(t))} nRNA_j,$$

$$nRNA_i \xrightarrow{P_d} 0,$$

$$nRNA_j \xrightarrow{P_d} 0.$$

More specifically, we simulated thousands of trajectories (15,000 s each) for each pair of genes for a given $\omega(r_{ij}(t))$ akin to the number of chromosomes within the experimental data. If the amount of nascent RNA for a gene was greater than 0 at the end of the trajectory, the gene was considered 'on' (Gene = 1), making our simulation data binary like the experimental data. Lastly, we incorporated the error due to the resolution of the experiment (resolution = 100 nm, 'Methods'). In total, using this numerical simulation approach, we are able to generate curves like *Figure 4A*, for a given coupling coefficient $\omega(r_{ij}(t))$, from the underlying spatiotemporal fluctuations of single genes in living cells. Importantly, the diffusive properties of active genes and the dwell time of nascent RNA are derived empirically from experimental data. Of the parameters described above, the coupling coefficient is the least well-determined and lacks an underlying mechanistic motivation at present.

Is it possible for a single function ($\omega(r_{ij}(t))$) to adequately reproduce the experimental results (*Figure 4A*)? To address this question, we iterated over many possible monotonically decreasing ($\omega(r_{ij}(t))$) functions. More specifically, we investigated all possible monotonically decreasing functions in 0.05 increments, with specific values for distances binned at a 200 nm resolution ('Methods,' *Figure 4E*). For each $\omega(r_{ij}(t))$, we quantified the correlation–distance curve for each gene pair and sought to find the one that was closest to *Figure 4A* ('Methods'). The best-performing $\omega(r_{ij}(t))$ is shown in *Figure 4E*, which resulted in the correlation–distance dependence in *Figure 4F*, demonstrating that a single general function can adequately describe this phenomenon at the level of the chromatin-tracing experiment.

With this dependence in hand, we are able to computationally remove processes that distort the correlation–distance relationship in an effort to uncover the 'true' observable degree of correlation for a given distance. The correlation–distance relationship in *Figure 4F* is also shown in *Figure 4G* with a new y-axis range to aid comparison. We started by simulating all pairs of genes as before but without the resolution error of the experiment with the determined $\omega(r_{ij}(t))$ (*Figure 4H*). Removing resolution error associated with light microscopy resulted in a slight increase in the correlation for the first distance bin, resulting in a 66% increase (*Figure 4H*). For all other distances, the degree of correlation was basically unchanged. We then simulated the system without resolution error and with a deterministic on time for each nascent RNA – each nascent RNA lasted exactly 800 s. We observed

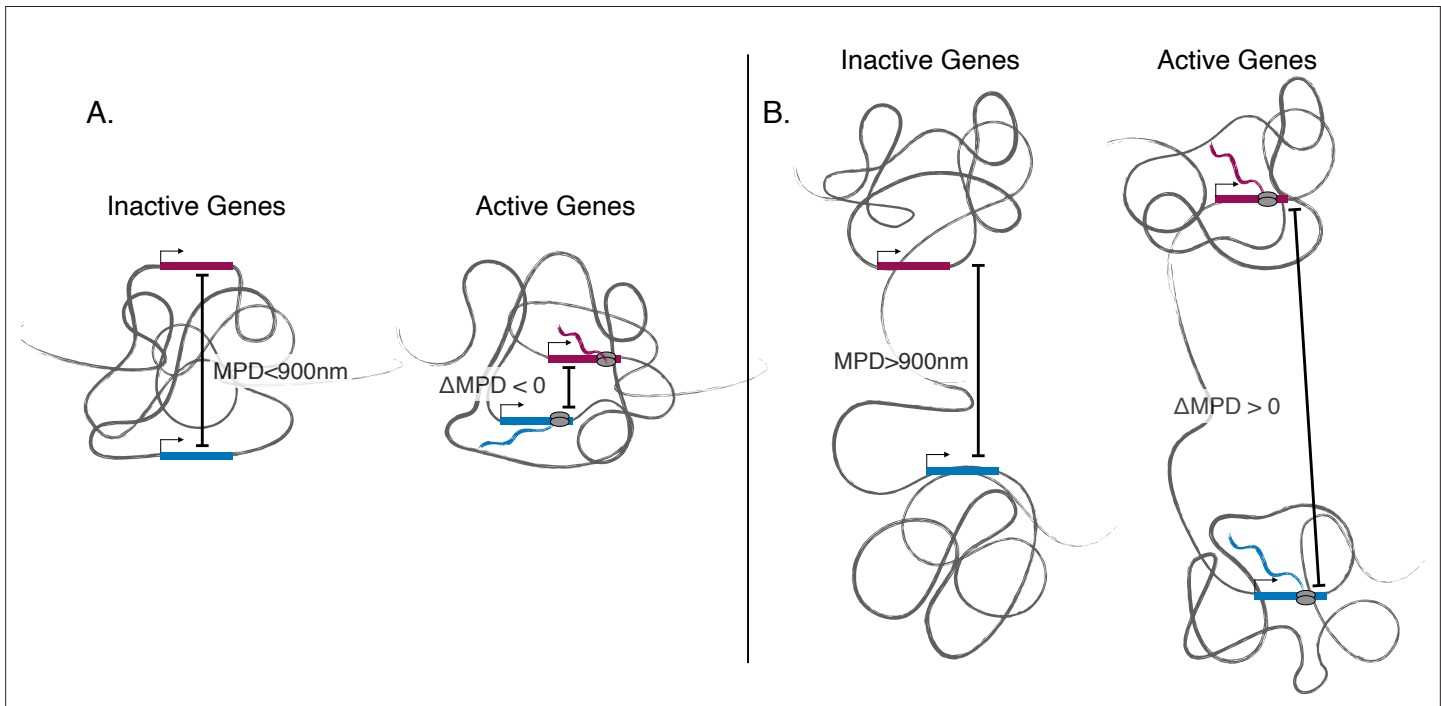

**Figure 5.** Illustration showing discovered phenomena. (**A**) An illustration showing the movement toward the same local centroid of genes separated by a smaller median physical distances (MPD). Here, the genes move toward the same local centroid and hence have a smaller MPD when active. (**B**) An illustration showing genes separated by a larger MPD. Here the genes still move toward their own local centroids when active but are arranged in such a way that they move away from each other when active.

a much greater increase across all distances with the first distance bin rising to 250% of its initial value (*Figure 4I*). Finally, we simulated the system removing resolution error, with deterministic on times, and without diffusion. Removing these three noise sources resulted in a large increase in correlation for lower distances and a slight decrease for larger distances (*Figure 4J*). This latter decrease is due to the correlated bursts at small distances not being able to diffuse to larger distance. For the first distance bin, the removal of all sources of error in fixed cell experiments leads to an ≈5-fold increase. The correlation is surprisingly high (≈ 0.3) and extends over a spatial distance of ≈ 400 nm. Overall, this analysis suggests that if one was able to monitor the distance between genes with high resolution and at time resolution where one could determine the exact start of each transcriptional burst, one should be able to see this true relationship – a clear direction for future pursuit.

## Discussion

By capitalizing upon the single-chromosomal nature of chromatin-tracing and nascent RNA smFISH data (*Su et al., 2020*), we discovered a variety of phenomena related to the coupling between transcription and higher order chromosome conformation. Specifically, fixed-cell analysis of chromatin conformation and activity coupled with live-cell analysis of transcription dynamics provides two features that are key to the analysis performed here: fluorescence microscopy reveals true physical distances and the variability across single cells. Leveraging these unique features, we find that (1) the chromatin around a gene is 'constrained' with transcription; (2) during a transcriptional burst genes are positioned toward the centroid of their surrounding chromatin; (3) transcriptional bursts cause promoters to move toward or away from each other depending on the MPD between them (These phenomena are illustrated within the simple model shown in *Figure 5*); (4) the distance between genes in individual cells is predictive of co-bursting; and (5) the lack of temporal information and limited imaging resolution greatly reduces the true distance–correlation relationship, with the predicted correlation coefficient of ~0.3 for a distance below 400 nm. This last finding relies on theoretical assumptions regarding chromatin mobility and the precise molecular nature of gene co-expression and awaits future experimental validation. At last, we should also note that more datasets from large-scale microscopy studies are likely on the way, where similar approaches to this study can be taken.

### Genes reposition upon transcriptional activation

Our finding that individual transcriptional bursts lead to the repositioning of genes and lower chromatin variability suggests the two phenomena could be linked. The traditional view of transcription influencing the dynamics of chromatin is that transcription leads to more 'open' and dynamic chromatin (*Babokhov et al., 2020*). While the traditional view has some empirical support (*Gu et al., 2018*), the exact opposite has been observed (*Germier et al., 2017*; *Nozaki et al., 2017*; *Nagashima et al., 2019*). Accepting the variability of distance distributions as a proxy for the motion of chromatin puts our observations in agreement with the latter. One possibility is once a gene is positioned toward the centroid of the surrounding chromatin, the confinement could be due to a new microenvironment. Another possibility – which we favor – is that the movement toward the centroid is a steric effect. Active genes recruit large megadalton complexes such as the pre-initiation complex and RNA polymerase II, which 'pushes' and confines the gene to a specific location due to the occluded volume effect. Our analysis thus suggests behavior consistent with the original factory model (genes reposition to a factory upon activation) and also the dynamic self-assembly model (genes assemble their own transcription factory). The order of events is key to distinguishing these alternatives, and these events are not resolved in the fixed-cell datasets analyzed here (*Cisse et al., 2013*; *Cho et al., 2016a*; *Cho et al., 2016b*; *Cho et al., 2018*; *Henninger et al., 2021*). Nevertheless, almost all of the ≈80 genes showed this behavior of repositioning and confinement, suggesting a general phenomenon, illustrating a fundamental aspect of transcription whose mechanistic details await additional study.

On a higher level, promoter–promoter distances (*Hsieh et al., 2020*) are clearly variable with individual transcriptional bursts and are likely important for understanding enhancer biology and other higher order functional assemblies. Considering the functional similarity between promoters and enhancers (*Kim and Shiekhattar, 2015*), we speculate that the rules of promoter–promoter interaction observed here may apply to enhancer–promoter interaction. In most cases, the distance change of promoters with transcription is small when compared to the MPD, but for MPD < 400 nm

a repositioning of 100 nm could be functionally relevant (*Figure 2C*; *Levo et al., 2022*; *Bohrer et al., 2021*; *Heist et al., 2019*; *Chen et al., 2018*; *Fukaya et al., 2016*) – putting the distances at the scale of enhancer–promoter communication (*Chen et al., 2018*). On the other hand, transcription factories have also been shown to be highly dynamic (*Cisse et al., 2013*; *Cho et al., 2018*; *Henninger et al., 2021*), raising the question of whether these dynamic promoter–promoter distances are linked to the dynamics of the factories (*Heist et al., 2019*). The unexpected finding that high MPD promoters tend to move away from each other with transcription suggests the possibility of specific locations for transcription, but this observation might also be used to explain specificity of enhancer–promoter interactions. Intriguingly, whether genes move toward or away from each is dependent upon ensemble chromatin organization, raising the possibility that genes are distributed according to chromatin organization and not genomic distance – given there is an underlying fitness advantage. Finally, it should be noted that for all these results described here there is a lack of temporal information, which obscures the cause and effect of these phenomena (just as we showed for the distance–correlation relationship). It therefore seems likely that these distance changes are likely more significant – a direction for future research.

## Genes in spatial proximity show high correlations in transcriptional activity: Interpreting $\phi \sim .3$

The hypothesis that genes in close spatial proximity are transcriptionally correlated has long persisted in the field despite conflicting data. Notable studies have taken advantage of single-cell RNA-seq and Hi-C data to disentangle the influence of genomic distance and physical distance on correlation with unclear results (*Sun and Zhang, 2019*; *Tarbier et al., 2020*). For example, while genes from the same (ensemble) topologically associated domain are more co-expressed, intra-chromosomal genes separated by similar genomic distances show essentially no difference in correlation with enrichments in contact frequency (*Tarbier et al., 2020*). The study of Sun et al. even found that the genomic distance is slightly more strongly correlated with co-expression than contact frequency (*Sun and Zhang, 2019*) – rightly explained away given the contact frequency was of a lower resolution with high error. Further, nascent RNA FISH found intra-chromosomal genes are not more correlated than when in trans (*Levesque and Raj, 2013*). Yet, single-cell imaging experiments coupled with detailed chromosomal perturbations have revealed spatial interactions that dictate a 'hierarchical' organization in multiple genes in response to stimulus (*Fanucchi et al., 2013*). Moreover, a recently proposed transcription factor activity 'gradient' model is a diffusion-based model that relies again on the spatial proximity of cis-acting regulatory elements, which might equally well be applied to promoter–promoter interactions (*Karr et al., 2022*). Overall, the hypothesis has persisted due to the intuitive mechanism even with the lack of definitive experimental demonstration.

Our results verify the null hypothesis and explain the negative results of previous single-cell studies. We found an enrichment in correlation for nascent RNA given that the genes are separated by a genomic distance of less than 2.5 Mb (*Figure 3A*). The fact that the average genomic distance between genes in the previous work was 3 Mb explains why enriched correlations were not seen at the nascent RNA level (*Levesque and Raj, 2013*). With our finding that the variability in MPD (or contact frequency) for a given a genomic distance is too low to disentangle these variables (*Figure 3D and E*), the defined enrichments in contact frequency for previous studies were likely quite minor in terms of producing a change in correlation (*Tarbier et al., 2020*). Utilizing the large amount of stochasticity in chromatin structure for individual chromosomes (*Finn and Misteli, 2019*) definitively shows the physical distance drives co-expression. This result is illustrated with the extremes: we observed an enrichment in correlation for genomic distances up to 10 Mb when the physical distance between genes was less than 200 nm on individual chromosomes, and very low correlations between genes separated by less than.5 Mb given that the physical distance was above 1200 nm (*Figure 4B*). In summary, our key finding is a correlation gradient with physical distance but not genomic distance.

The lack of temporal data and the spatial resolution limits of the chromatin-tracing methodology greatly obscures both the 'true' transcriptional correlation between spatially proximal genes and also the length scale over which transcriptional correlation is measured. The reasons for this reduced correlation are obvious: both the position and the activity status of genes vary randomly. One can imagine, for example, genes that were far apart at activation and then diffused together and vice versa. Correcting for this behavior requires assumptions about chromatin mobility and also utilization

of live-cell nascent RNA data. We predict that if one were able to measure the distances between genes at the initiation of the transcriptional bursts, one should obtain a correlation of ~0.3 if the distance between the promoters of the genes is less than 400 nm. Intriguingly, this level of correlation has been reported between the mRNA levels of adjacent genes in yeast but was attributed to DNA supercoiling (*Patel et al., 2022*). Considering the shorter lifetimes of mRNA in yeast, this correlation may be comparable to the nascent RNA in humans. Furthermore, other live-cell studies have seen correlated bursts between spatially proximal genes (in trans and cis), but did not specifically investigate this as a function of the physical distance between the genes or account for the variable on times (*Fukaya et al., 2016*; *Lim et al., 2018*; *Heist et al., 2019*; *Levo et al., 2022*) – finding enrichments in correlation similar to the uncorrected curve (*Figure 4A*; *Levo et al., 2022*). The enrichment in co-bursting for genes separated by <400 nm suggests the working distance of the underlying mechanism is not direct contact. Exactly what mechanism leads to these general correlations is still unknown; however, these results are consistent with the idea of enhancers coordinating transcription with working distances of hundreds of nm (*Fukaya et al., 2016*; *Lim et al., 2018*; *Heist et al., 2019*; *Levo et al., 2022*; *Bohrer et al., 2021*).

The analysis suggests co-expression is a general property of the system, that is, unrelated genes show correlated bursts with each other when in spatial proximity. This transcriptional correlation would then be an unavoidable emergent behavior due to the physicality of the system. Hence, the appearance of correlated bursts may not suggest a specific regulatory mechanism. Stated another way: we hypothesize that the physical distance between the vast majority of genes arises from the physical constraints of the nucleus and DNA and is not indicative of a biologically functional relationship requiring coordinated expression conferred by that proximity. Support for this hypothesis comes from the observation that disrupting genomic clusters of metabolic genes such as the *GAL* genes in yeast have no measurable impact on fitness (*Lang and Botstein, 2011*). Of course, there are certainly instances where coordinated co-expression conferred by spatial proximity is important, for example, in the segmentation clock genes *her1* and *her7* located on the same chromosome and separated by 12 kb (*Zinani et al., 2021*). The corollary to our hypothesis is that one can look for deviations from the $\phi \sim .3$ to identify bona fide regulatory relationships. Thus, we establish a theoretical benchmark that can be used in future studies.

Lastly, we should note here that we consider this methodology a first theoretical step due to the lack of information about the underlying mechanisms on the chromosomal scale. Therefore, future work should be the adaption of more complicated chromatin polymer models to refine our understanding of this phenomena – of special note are those that explicitly model the links between chromatin organizations and their influence on transcription regulation (*Brackley et al., 2021*). These future models will likely need to explicitly model the underlying processes (like loop extrusion) to capture the variability in chromatin structure and dynamics whose specifics are likely to emerge in future studies – either validating or suggesting modifications to our approach above.

## Methods

### Expected MPD and genomic distance

To determine the expected MPD for a given genomic distance, we simply calculated the average MPD for each specific genomic distance. For example, to determine the expected MPD for a genomic distance of 50 kb, we quantified the average MPD between all loci separated by 50 kb.

To determine the expected genomic distance for a given MPD, we used the same curve and found the genomic distance with the closest average MPD. For example, say the MPD between two loci is 500 nm, using the previously quantified curve, the expected genomic distance is the genomic distance whose average MPD is closest to 500 nm.

### Correlations between genes

When quantifying the correlations between a pair of genes (aka. whether they were on or off, 1 or 0), we quantified the $\phi$ coefficient (used for binary data):

$$\phi = \frac{n_{11}n_{00} - n_{10}n_{01}}{\sqrt{(n_{11}+n_{10})(n_{11}+n_{01})(n_{00}+n_{10})(n_{00}+n_{01})}},$$

where $n_{11}$ is the number of observations where both genes are active and $n_{10}$ is the number of observations where the first gene is on and the second is off, etc. Here, we should state that $\phi$ is equivalent to the Pearson correlation coefficient and the Spearman correlation coefficient for this data due to a gene's transcription state being either 1 or 0 – that is, on (1) or off (0).

## Determining $P_i^{tot}$

To determine the bursting propensity for each gene, we first conducted many different simulations with $P_i^{tot}$ values ranging from 0 to 0.05 with our set nRNA decay rate. For each propensity, we simulated 2000 trajectories (15,000 s each). Then, with the last timepoints of each trajectory, we classified the gene as being either 'on' or 'off' – if the gene's nRNA was greater than zero, the gene was classified as 'on' (aka 1). We then simply created a lookup table with the average number of 'on' states vs. the bursting propensity. To determine a genes specific propensity, we simply calculated the average number of 'on' state with the experimental data and found the closest match within the lookup table.

## Modeling co-transcriptional bursts

To account for co-expression for a pair of genes, we modeled nascent RNA production as coming either from a co-burst or from an individual burst:

$$P_i^{tot} = P_{ij}(r_{ij}(t)) + P_i(r_{ij}(t)), \tag{1}$$

$$P_j^{tot} = P_{ij}(r_{ij}(t)) + P_j(r_{ij}(t)). \tag{2}$$

Here, $P_{ij}(r_{ij})$ is the probability of a transcriptional co-burst per second given the distance between the two genes, $P_i(r_{ij})$ is the probability of an individual burst per second given the distance, and $r_{ij}(t)$ was determined beforehand utilizing the above Langevin equation specific for that gene pair ('Methods).

The fact that genes have different expression levels limits the values of $P_{ij}(r_{ij}(t))$. Arranging the pair of genes so that $P_i^{tot} < P_j^{tot}$, the maximum value that $P_{ij}(r_{ij}(t))$ can be is $P_i^{tot}$ – or else $P_i(r_{ij}(t))$ would have to be negative. With this, we can then rewrite the above as the following:

$$P_{ij}(r_{ij}(t)) = \omega(r_{ij}(t)) \times P_i^{tot}, \tag{3}$$

$$P_i(r_{ij}(t)) = P_i^{tot} - \omega(r_{ij}(t)) \times P_i^{tot}, \tag{4}$$

$$P_j(r_{ij}(t)) = P_j^{tot} - \omega(r_{ij}(t)) \times P_i^{tot}, \tag{5}$$

where $\omega(r_{ij}(t))$ is a function of distance between the genes and ranges between 0 and 1. $\omega(r_{ij}(t))$ is the proportion of gene i's transcriptional bursts that are co-bursts at each distance; if the expression levels of the two genes are approximately equal, $\omega(r_{ij})$ is equal to the proportion of bursts that are co-bursts at a given distance for both genes.

Overall, with a single function ($\omega(r_{ij}(t))$), we modeled all pairs of genes with the following stochastic reactions utilizing the Gillespie algorithm (*Gillespie, 1977*):

$$0 \xrightarrow{P_{ij}(r_{ij}(t))} nRNA_i + nRNA_j,$$

$$0 \xrightarrow{P_i(r_{ij}(t))} nRNA_i,$$

$$0 \xrightarrow{P_j(r_{ij}(t))} nRNA_j,$$

$$nRNA_i \xrightarrow{P_d} 0,$$

$$nRNA_j \xrightarrow{P_d} 0.$$

## Incorporating resolution error

The resolution of the experimental data was previously quantified in the work of *Su et al., 2020*, and the resolution of each chromosomal segment was determined with approximately 100 nm resolution. The 3D resolution error is not Gaussian due to the Pythagorean theorem and was determined by *Churchman et al., 2006*. Therefore, for our case, the error must be applied to all three dimensions independently – similar to in Su et al. To do this, with the 'true distance' from the Langevin simulation we randomly decompose the distance into three dimensions – so that the distances along each dimension satisfy the Pythagorean theorem. We then added two random variables of Gaussian noise

with standard deviations of 100 nm (one for each loci), generating a new displacement for each dimension with localization error. Lastly, we took the displacements along each dimension with the error and quantified the distance in 3D using the Pythagorean theorem.

## Quantifying best $\omega(r_{ij})$

To determine the $\omega$ that captures the behavior of the experimental data, we first generated a large number of unique monotonically decreasing functions. This was first done in 0.1 iterations and with a distance binning of 200 nm. For example, $\omega^1(r_{ij}) = [.1, 0, 0, 0, 0, 0, 0, 0, 0, 0, 0, 0, 0, 0, 0, 0]$ means genes that are within 200 nm of each other (first number in array) have the value 0.1, and the rest of the distances have the value 0. We would then iterate and produce the next $\omega$, $\omega^2(r_{ij}) = [.1, .1, 0, 0, 0, 0, 0, 0, 0, 0, 0, 0, 0, 0, 0, 0]$, etc. We then simulated a large number of trajectories for all gene pairs according to the model in the main text with each function. We then quantified the error between each $\omega$'s distance–correlation relationship and the experimental data with the following:

$$Error(\omega^k) = \sum_i \sum_j \sum_r |\phi_{ij}^{\omega^k}(r) - \phi_{ij}^{exp}(r)|,$$

where $\phi_{ij}^{\omega^k}(r)$ is the correlation for the gene pair $ij$ given that the observed distances were within the distance bin $r$ (200 nm for each bin) and $\phi_{ij}^{exp}(r)$ is the correlation for the experimental data for that gene pair. Once we found the $\omega$ that resulted in the minimum error was found, we then varied the values for distance bins below 1000 nm by plus or minus 0.05. We then quantified the error again to result in the best-fit function shown in the main text.

## Mean squared displacement (MSD)

We quantified the motion of the *TFF1* gene utilizing the multiple allele data from *Rodriguez et al., 2019*. This live-cell data provided the 2D coordinates of active alleles over extended periods of time, allowing us to monitor the motion of chromatin over a timescale longer than the on time of a gene. To account for the movement of the cell over these long periods, we monitored the motion of one tagged allele relative to another. We then quantified the MSD for a given time (Δt): MSD(Δt)=<[R(t)-R(t-Δt)]². Where R(t) is the position of an allele relative to another, and the arrows are the ensemble average and over all measured trajectories and times.

## Modeling distance diffusion

To model the distance between two chromosomal loci, we utilized the following Langevin equation:

$$\frac{dr_{ij}}{dt} = -\frac{1}{\gamma_{ij}} \frac{\partial V_{ij}(r_{ij})}{\partial r_{ij}} + \sqrt{2D} \times g(t).$$

Here, $r_{ij}$ is the distance between genes $i$ and $j$, $V_{ij}(r_{ij})$ is the potential (specific to that gene pair, described below), $\gamma_{ij}$ is a constant specific for that gene pair, and the last term $\sqrt{2D} \times g(t)$ accounts for the Brownian motion with the determined diffusion coefficient – if the potential is a constant independent of distance, $r_{ij}$ will exhibit Brownian motion. For each gene pair, we empirically determined a $\frac{1}{\gamma_{ij}} \frac{\partial V_{ij}(r_{ij})}{\partial r_{ij}}$ that 'biases' the distance's motion so the steady-state distribution matches the empirically determined distance distribution (corrected for the resolution of the experiment) – this accounts for the genes being on the same chromosome.

The equivalent Fokker–Planck equation is

$$\frac{\partial P_{ij}(r_{ij},t)}{\partial t} = \frac{1}{\gamma_{ij}} \frac{\partial}{\partial r_{ij}} [\frac{\partial V_{ij}(r_{ij})}{\partial r_{ij}} P_{ij}(r_{ij}, t)] + D \times \frac{\partial^2 P(r_{ij},t)}{\partial r_{ij}^2},$$

where the initial condition is dropped for simplicity and $P_{ij}(r_{ij}, t)$ is the probability distribution to have a distance $r_{ij}$ at time $t$ specific to that gene pair. We then set the left hand of the equation equal to zero, defining the steady-state distance distribution ($P_{ij}^s(r_{ij})$). The equation then becomes

$$\frac{1}{\gamma_{ij}} \frac{\partial V_{ij}(r_{ij})}{\partial r_{ij}} P_{ij}^s(r_{ij}) + D \times \frac{\partial P_{ij}^s(r_{ij})}{\partial r_{ij}} = 0$$

with the solution

$$P_{ij}^s(r_{ij}) = C_{ij} \times exp(-\frac{V_{ij}(r_{ij})}{\gamma_{ij}D}),$$

where $C_{ij}$ is a normalization constant.

From the experimental data, we can empirically determine $P_{ij}^s(r_{ij})$. To do this, we took the naturally observed distance distribution and performed a deconvolution with the resolution distribution. This provided us with $P_{ij}^s(r_{ij})$ minus the resolution error, and we can therefore solve for the potential with

$$\frac{V_{ij}(r_{ij})}{\gamma_{ij}} = D[ln(C_{ij}) - ln(P_{ij}^s(r_{ij}))]$$

With this we can then simulate the Langevin equation with the Euler–Maruyama method, which results in the proper steady-state distribution with the approximate diffusion coefficient.

## Acknowledgements

This work would not have been possible without the computational resources of the NIH HPC Biowulf cluster (http://hpc.nih.gov). We would also like to thank the members of the Larson lab for their input, especially Dr. Nadezda Fursova. Additionally, we would like to thank Dr. Alexander Englert for the useful discussions.

## Additional information

### Funding

| Funder | Grant reference number | Author |
| --- | --- | --- |
| National Institutes of Health | 1ZIABC011383-11 | Christopher H Bohrer Daniel R Larson |

The funders had no role in study design, data collection and interpretation, or the decision to submit the work for publication.

### Author contributions

Christopher H Bohrer, Conceptualization, Data curation, Software, Formal analysis, Validation, Investigation, Visualization, Methodology, Writing - original draft, Writing – review and editing; Daniel R Larson, Conceptualization, Resources, Supervision, Funding acquisition, Investigation, Project administration, Writing – review and editing

### Author ORCIDs

Christopher H Bohrer (iD) http://orcid.org/0000-0002-2017-7933
Daniel R Larson (iD) http://orcid.org/0000-0001-9253-3055

### Decision letter and Author response

Decision letter https://doi.org/10.7554/eLife.81861.sa1
Author response https://doi.org/10.7554/eLife.81861.sa2

## Additional files

### Supplementary files
• MDAR checklist

### Data availability

The current manuscript is a computational study. Analysis code and modeling code are included in GitHub https://github.com/CHB-Bohrer/co-bursting (copy archived at swh:1:rev:6f85565959fccb790bfd448831d8211be6f5a57e).

The following previously published dataset was used:

| Author(s) | Year | Dataset title | Dataset URL | Database and Identifier |
|---|---|---|---|---|
| J-H Su, Zheng P, Kinrot SS, Bintu B, Zhuang X | 2020 | Genome-Scale Imaging of the 3D Organization and Transcriptional Activity of Chromatin (chromosome21.tsv) | https://doi.org/10.5281/zenodo.3928890 | Zenodo, 10.5281/zenodo.3928890 |

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

## Appendix 1

### H3K27ac analysis

To quantify the density of H3K27ac within each corresponding 50 kb segment of Chr21 in IMR90 cells, we utilized the Chip-seq data from the Bing Ren Lab at UCSD. https://www.encodeproject.org/experiments/ENCSR002YRE/. More specifically, we quantified the average number of reads within each 50 kb segment from two biological repeats – this was done using the software packages Samtools and deepTools. We then normalized the reads by dividing by the sum, allowing us to understand these values in relation to the whole – this is shown in *Appendix 1—figure 1A*. To understand whether there is a dependence upon the transcription-induced repositioning of the genes based on the H3K27ac signal, we then partitioned each locus into one of four groups (low, med, high, very high, *Appendix 1—figure 1A*) and quantified the repositioning based off of the H3K27ac density (*Appendix 1—figure 1B*, colors).

### Method specifics for single-locus diffusion

To investigate the diffusive behavior of transcriptionally active genes that were tagged at a single allele, we utilized the live-cell microscopy data for four different genes (*MYH9, RAB7A, CANX, SLCA1*) from Wan et al. Of note, this data is different from the multiallele diffusion analysis within the main text in that there was no internal nuclear reference point to correct for cellular movement over these long timescales. Still, in order to try and correct for the cellular movement we segmented the nucleus using the background GFP signal, resulting in a binary image of which pixels belonged to the nucleus and which did not. We then utilized the center of mass of the nucleus of the cell to adjust the diffusive trajectory within that cell.

### Simulation for single- and double-locus diffusion

To understand how the diffusion of the single-allele genes relate to the multiallele *TFF1* data within the main text, we sought to utilize a simple 2D random diffusion model to simulate the diffusive behavior of the two. This is important as the diffusion coefficient we seek to capture for the model is the distance between two different chromosomal loci. To do this, we simulated a simple random 2D walks consisting of either one particle or two particles with 1000 individual trajectories each with a time of 10,000 s. Each of the particles was simulated with a diffusion coefficient approximately equal to that of *RAB7A* ($D = .1e-3 \mu m^2/sec$). When we quantified the diffusion coefficients of the single particles by fitting the 2D MSDs of the simulated data it resulted in the proper diffusion coefficient (*Appendix 1—figure 5B*). Then when we quantified the diffusion of the simulations with two particles – taking the distance of one relative to the other, similar to that of *TFF*1 – the MSD resulted in a coefficient approximately double ($D = .2e-3 \mu m^2/sec$), suggesting that the diffusion of the single-locus data is more similar to the *TFF*1.

### Specifics on statistics

Bootstrapping methodology

The bootstrapping shown within the box plots of the main text was calculated utilizing the Python plotting software seaborn, with the pointplot function. More specifically, the estimator was the Python software numpy's mean function and the number of bootstraps was 1000. From these, the standard error of mean was quantified and displayed using the seaborn pointplot function.

Statistical significance for box plots

The significance quantified for the data shown within the boxplots is defined as having a p-value < 0.01 determined using a t-test. The specific software used to perform the t-test was the Python software SciPy with the stats package and the specific function ttest-ind.

Statistical significance for average correlation

To quantify if the average correlation values were themselves correlated along a specific dimension (*Figures 3 and 4*), the Python software SciPy was used with the stats package and the spearmanr function. The spearmanr function quantifies the monotonicity between two datasets and also produces a p-value that is equivalent to 'the probability of an uncorrelated system producing datasets that have the same Spearman correlation coefficient.' We, therefore, defined a significant correlation along a dimension (for the average correlation values) those that resulted in a p-value <0.01.

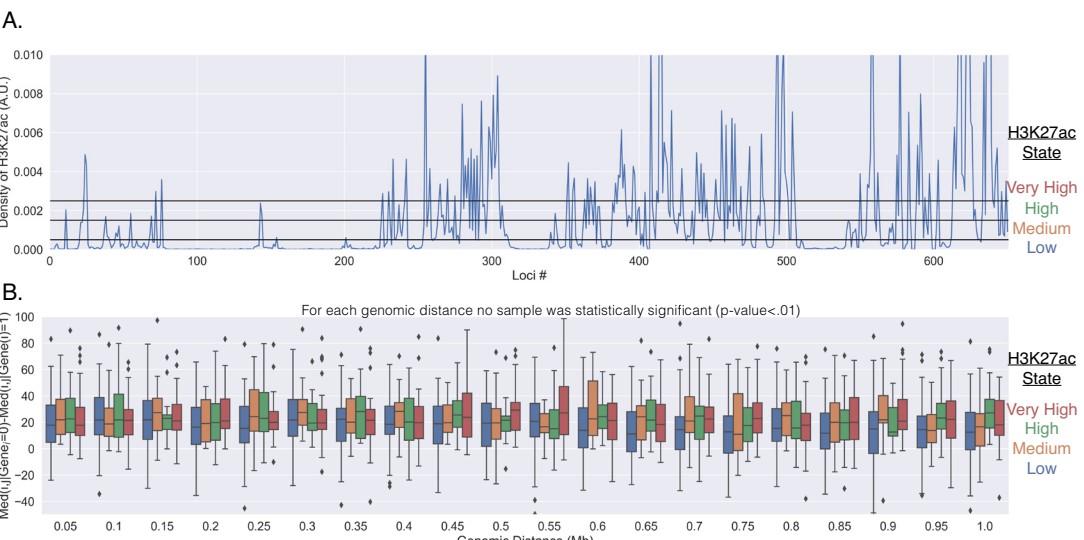

**Appendix 1—figure 1.** Genes do not reposition for enhancer activation. (**A**) The normalized number of reads within the corresponding 50 kb segment of chromosome 21. The reads were normalized by the total number of reads from the 651 chromosomal segments. The black horizontal lines show how the H3K27ac signal was partitioned into each group (low, medium, high, very high). (**B**) The difference in the median physical distance (MPD) between loci $i$ and $j$, given the transcription state of the investigated gene located within loci $i$. This difference is shown as a function of the genomic distance between the loci and was partitioned based off of the the H3K27ac state of loci $j$ (the different colors). For each genomic distance bin, t-tests were performed on the various pairs, and none were found to be significant (p-value < 0.01).

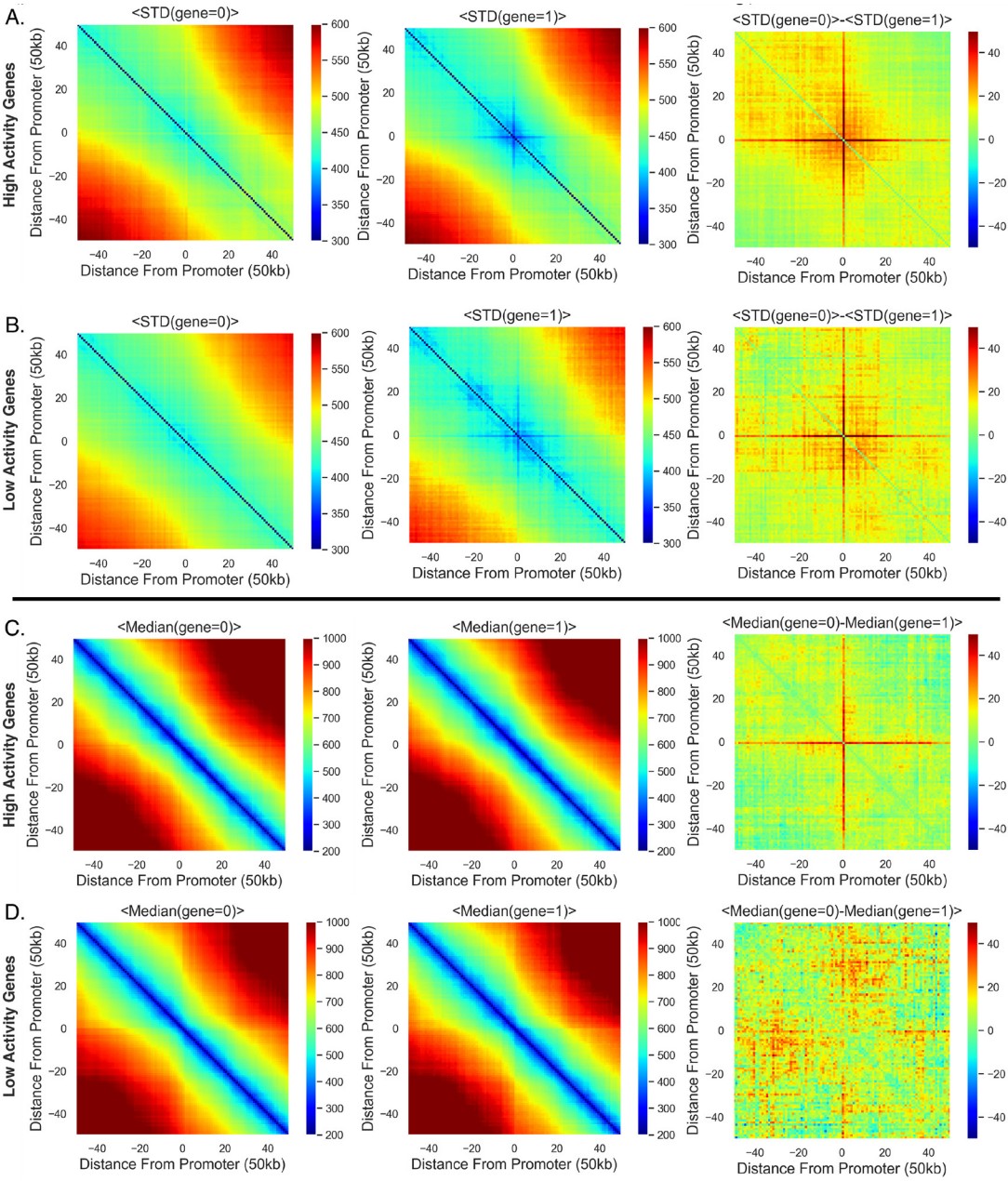

**Appendix 1—figure 2.** High-activity genes are more constrained with transcription and show a stronger repositioning trend. (**A**) The average standard deviation (over all high-activity genes) given the transcription state of the gene, and the difference between the average standard deviations with the different transcription state. (**B**) Same as (**A**) but for the low-activity genes. (**C**) The same as (**A**) but for the average median distances for the high-activity genes. (**D**) Same as (**C**) but for the low-activity genes.

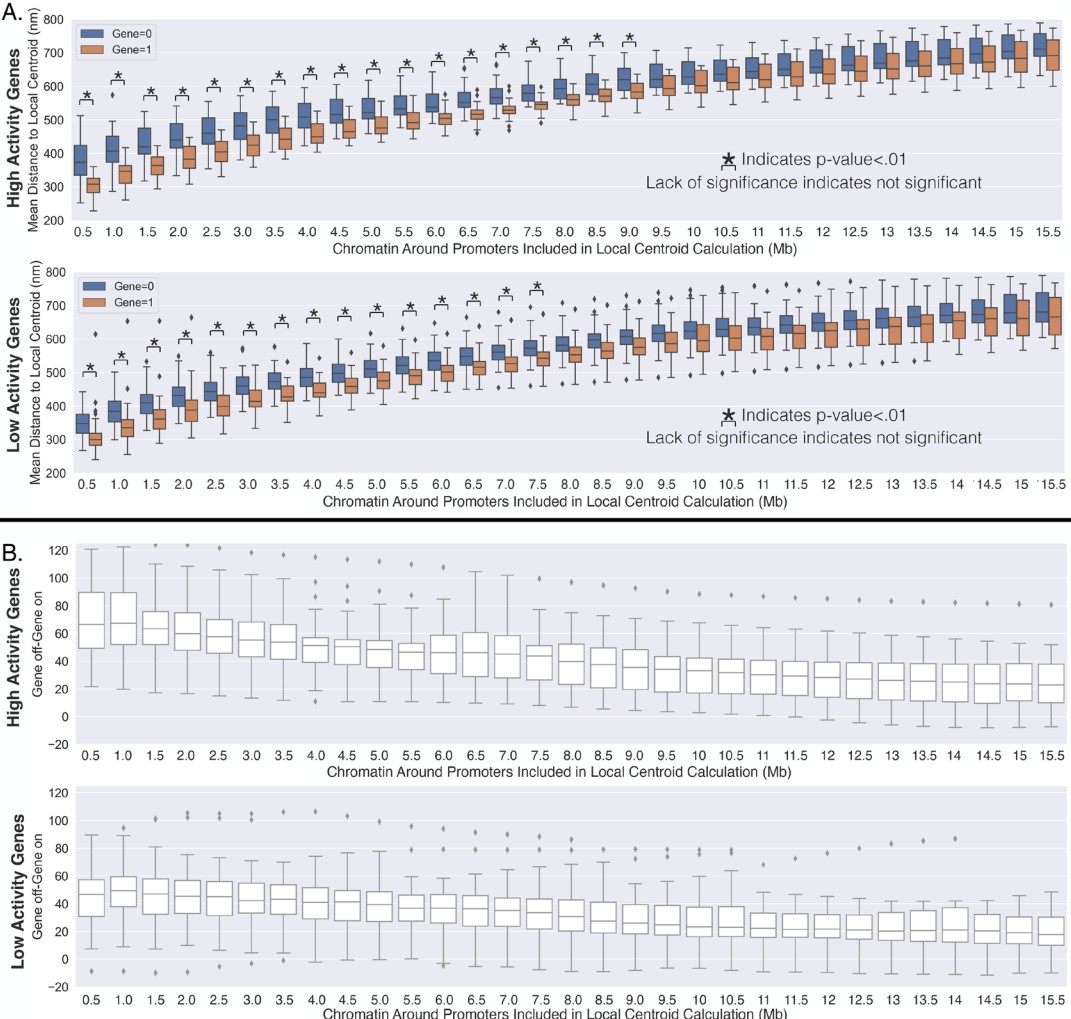

**Appendix 1—figure 3.** High-activity genes travel a farther distance toward the local centroid with transcription activation. (**A**) The average distance to the local centroid as a function of the amount of chromatin included within the centroid calculation. This is calculated for high-activity genes (first row) and the low-activity genes (second row). Significance was defined as a p-value <0.01 with a t-test (Appendix 1). (**B**) The change in the average distance to the local centroid with transcription activation on a gene-by-gene basis (similar to the main text). The first row is for the high-activity genes, and the second row is for the low-activity genes.

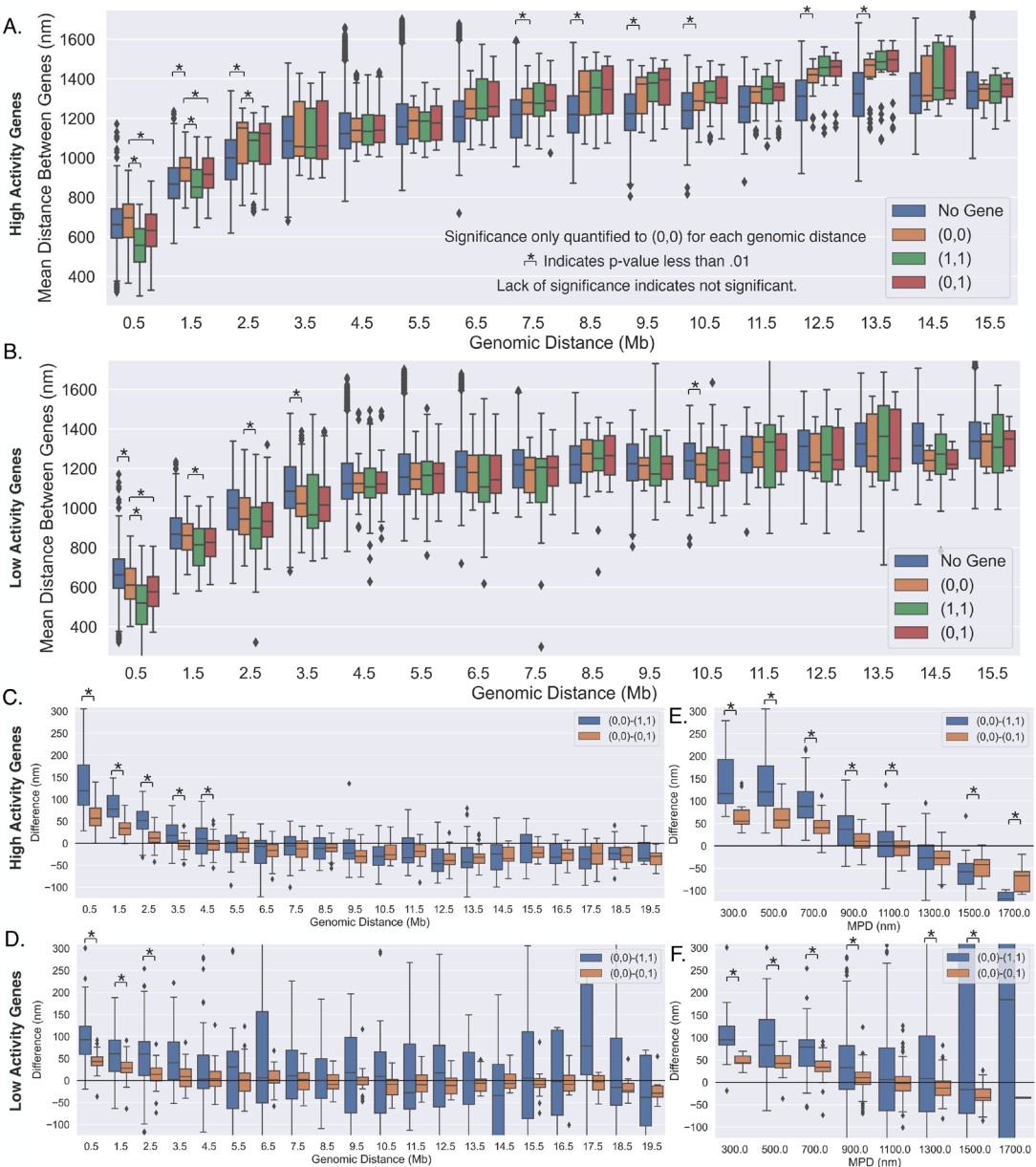

**Appendix 1—figure 4.** Pairs of high-activity genes move greater distances, toward or away from each other, depending upon transcription. (**A**) The average distances between pairs of high-activity genes (depending upon transcription state) as a function of the genomic distance. (**B**) The average distances between pairs of low-activity genes (depending upon transcription state) as a function of the genomic distance. (**C**) The difference between the scenarios shown in (**A**), showing the difference in mean distance on a gene pair by gene pair basis, and a black line is shown to aid in the visualization of zero. (**D**) The difference between the scenarios shown in (**B**), showing the difference in mean distance on a gene pair by gene pair basis, and again a black line is shown to aid in the visualization of zero. (**E**) The same analysis in (**C**), but as a function of the median physical distance (MPD) between the high-activity genes. (**F**) The same analysis in (**D**), but as a function of the MPD between the low-activity genes. For all subplots: significance was defined as a p-value <0.01 with a t-test (Appendix 1).

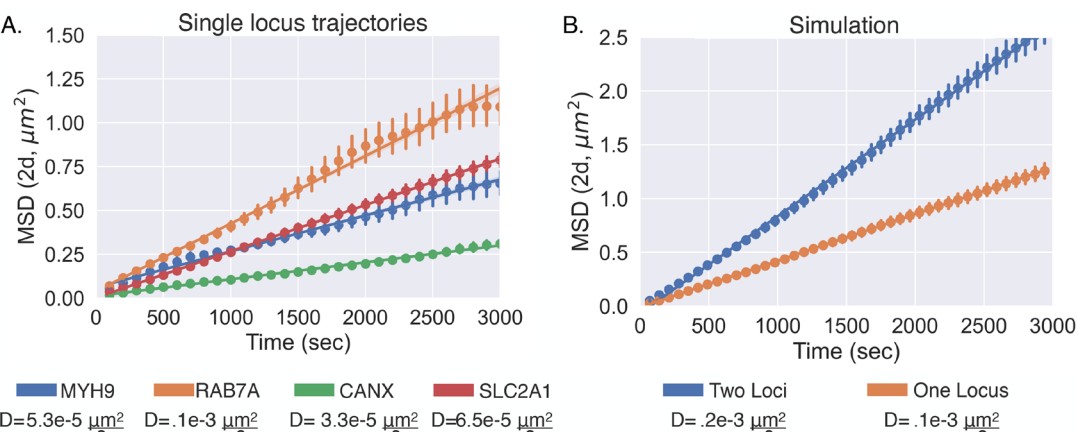

**Appendix 1—figure 5.** Diffusive behavior of single-allele-tagged genes. (**A**) The mean squared displacement of four different genes with the fitted lines and error bars showing individual 95% confidence intervals (Appendix 1). The diffusion coefficients are listed under each gene for reference. (**B**) The mean square displacement from the simple diffusion simulation (see Appendix 1) illustrates how the diffusion coefficient increases when considering the distance of one locus relative to the other. Again, there are the fitted lines and error bars showing individual 95% confidence intervals (Appendix 1).

