## [Editor Report]

In this article, Bohrer and Larson revisit previously published imaging datasets in order to tackle a long-standing question in modern genome biology: does the physical proximity of transcribed genes correlate with their co-expression? The authors provide convincing evidence to deduce that when a pair of loci are brought within sufficiently low physical 3D proximity (unrelated to their genomic distance) they are more likely than expected to be co-expressed. This is a result of potentially fundamental importance.

---

## [Decision Letter]

**Decision letter after peer review:**

Thank you for submitting your article "Synthetic analysis of chromatin tracing and live-cell imaging indicates pervasive spatial coupling between genes" for consideration by *eLife*. Your article has been reviewed by 2 peer reviewers, and the evaluation has been overseen by a Reviewing Editor and James Manley as the Senior Editor. The following individuals involved in review of your submission have agreed to reveal their identity: Robert A Coleman (Reviewer #1); Argyris Papantonis (Reviewer #2).

Using spatial analysis and modeling, the authors have impressively extended the findings of Su et. al, Cell 2020, who generated the analyzed dataset. A number of important concepts were explored including (1) do genes re-position upon activation and (2) can spatial proximity be correlated with transcriptional co-regulation. In general the authors conclusions are supported by their findings and should provide a blueprint for analysis of additional related big imaging datasets in the future.

Both reviewers find the manuscript important and valuable but have suggestions for improvement of clarity and analyses. These include:

Statistical analysis of the significance of the data needs to be done.

The writing is dense and should be made more readable, less jargon and details that would be more appropriate in the methods. A graphic image would help.

The authors should explore stratifying ON states in high and low to see if additional insights can be extracted.

*Reviewer #1 (Recommendations for the authors):*

(1) The authors should determine the statistical significance of their findings for figures 1 and 2, along with a thorough description of their bootstrapping and statistical analysis methods in the methods section.

(2) If possible, for Figure 1, it would be highly insightful to see whether known enhancer elements are moving closer to promoters of target genes during transcriptions as a comparison to their existing promoter-promoter data.

(3) An extension of the author's findings would be that histone marks associated with transcriptional activity (e.g. H3K27ac) would be enriched in chromatin loci that are in close proximity to the promoter when the gene is on. As a control, chromatin loci containing histone marks associated with gene activity (e.g. H3K27me3) would not move much between the on and off state. In other words, for a locus that is closest in proximity to a promoter, it would be very beneficial to measure the degree of H3K27ac (e.g. a mark of enhancer activity) compared to other surrounding loci of greater physical distance. ChIP-seq datasets for a variety of histone marks are available for the authors to perform this analysis.

(4) The changes in MPD stated in Figure 1I seem to be confined to a small region within 50Kb. How would the data look in Figure 1J/K if smaller bin sizes (e.g. 50Kb) were chosen instead of 500Kb?

(5) Given the authors findings on chromosome dynamics obscuring true correlation, it would be helpful to see if other datasets exist that measure the diffusion of a locus when the gene is turned on as comparison to the TFF1 mobility. Can authors compare the diffusion of MS2- labeled intronic sequences where they have a much larger dataset to draw upon? How does this mobility compare with dCas9 measurements examining diffusion of loci that presumably aren't transcribing.

(6) Representation of figures should be improved for increased clarity (e.g. Figures 1J/K, 2, 3A-C, 4 have data cutoff).

(7) As a way of orienting a non-specialist reader, it might be very helpful to see a representative tracing map of chromatin/promoter loci centroid repositioning upon transcriptional activity.

(8) One way to increase the general impact of this type of study is to lean into the fact (e.g. further emphasize in the text) that more big imaging datasets are on the way. As such, this study is a good example that re-examining publicly available datasets in a new way can lead to fundamental new insights or answers to long standing questions in the field.

*Reviewer #2 (Recommendations for the authors):*

I think that the following points, if addressed, can further strengthen a very interesting manuscript.

– The analysis is now confined to "on" (1) and "off" gene expression (0) states. I am wondering if the data provide the possibility to stratify the "on" genes in at least "low" and "high" categories and repeat analysis. These categories could reflect high/low FISH signal and/or high/low bursting frequency in the population (something the authors try to incorporate via their live-cell data; see my other comment below).

– For the analysis in Figure 3, contact frequency is deduced using high-resolution Hi-C data (not clear to me which and at which resolution to match that of the imaging). However, it is now well understood that Hi-C is generally depleted from promoter-promoter contacts, and that promoter-capture "C" data can prove tricky to quantify and can carry biases. On the other hand, Micro-C data would work very well here and might even reconcile the imaging with "C" technologies.

– Finally, regarding the (otherwise commendable) effort to generate a model that allows them to "merge" live-cell with fixed-cell data, the authors (i) make a number of assumptions that can be debated, and (ii) use a perhaps too parsimonious way to model chromatin behaviour. As regards (i), a key example is the generalisation of parameters based on analysis of a single locus, TFF1. Similarly the generalisation of ~13 min time for nascent RNA decay probability for all genes based on the MS2 FISH data from ref. 49 is not clear to me. As regards (ii), I think we must acknowledge that in silicon models of chromatin (also linked to transcription output, like the recent Brackley et al., 2021 Nat Commun paper by the Marenduzzo lab) from a number of labs (Mirny, Marenduzzo, Nicodemi, etc.) are growing more and more complex and approximate chromatin and gene expression behaviour evermore accurately. The model employed here is empirically tuned to match aspects of the data, but does not simulate many of the mechanisms known to work on chromatin (like extrusion, which the authors specifically also refer to). I would also like to note that this part of the paper is the least approachable to the average reader, leaves some concepts without any explanation and would benefit from some rewriting; the Results should describe the essence of the model and its key assumptions clearly, and the more complicated math and jargon should be detailed in the Methods, in my view.

– Last, I would like to note that the 400 nm cutoff deduced here is not at all unreasonable given previous data on "transcription factory" sizes (diameters between 85-250 nm) and the resolution of the analysed data. Mention of these in the Discussion could strengthen the postulation by the authors. Their statement reading "enrichment in co-bursting for genes separated by < 622 nm suggests the working distance of the underlying mechanism is not direct contact" should be accordingly tuned. Also, a comparison to the sizes of "condensates" would be welcome. Nonetheless, I was very happy to see that the manuscript offers a very balanced interpretation of results, previous work, and existing caveats, and was very nice to read overall.

---

## [Author Response]

Both reviewers find the manuscript important and valuable but have suggestions for improvement of clarity and analyses. These include:Statistical analysis of the significance of the data needs to be done.

We added statistical significance and further statistics where comparisons are needed. We show these specific modifications in the address to the first reviewer.

The writing is dense and should be made more readable, less jargon and details that would be more appropriate in the methods.

Here we have mainly modified the modeling section of the manuscript, which was referred to as the “least approachable to the average reader” by reviewer 2 - moving the “jargon” to the methods.

A graphic image would help.

We have added a graphic image to the discussion to illustrate the central findings.

The authors should explore stratifying ON states in high and low to see if additional insights can be extracted.

Given the available data, we were not able to partition the genes based on the “intensity” of the smFISH signal. Still, we have performed the same analyses on high and low activity genes when appropriate, using the fraction of genes in the ON state. The results of partitioning the genes clearly suggest a dependence on activity and have led to the addition of text as well. We show these specific modifications in the address to the second reviewer.

Reviewer #1 (Recommendations for the authors):(1) The authors should determine the statistical significance of their findings for figures 1 and 2, along with a thorough description of their bootstrapping and statistical analysis methods in the methods section.

We have now added statistical significance to Figures 1 and 2 and have added further details to the new Supporting Material discussing the bootstrapping and statistical methods (See Supporting Material, Lines 40:56).

Specifics on Statistics Bootstrapping Methodology

“The bootstrapping shown within the box plots of the main text was calculated utilizing the python plotting software seaborn, with the pointplot function. More specifically, the estimator was the python software numpy's mean function and the number of bootstraps was 1000. From these, the standard error of mean was quantified and displayed using the seaborn pointplot function.”

Statistical Significance for Box Plots

“The significance quantified for the data shown within the boxplots is defined as having a p-value <.01 determined using a t-test. The specific software used to perform the t-test was the python software scipy with the stats package and the specific function ttest-ind.”

Statistical Significance for Average Correlation

“To quantify if the average correlation values were themselves correlated along a specific dimension (Main Text Figures 3 and 4), the python software scipy was used with the stats package and the spearmanr function. The spearmanr function quantifies the monotonicity between two datasets and also produces a p-value which is equivalent to "the probability of an uncorrelated system producing datasets that have the same Spearman correlation coefficient." We, therefore, defined a significant correlation along a dimension (for the average correlation values) those that resulted in a p-value less than.01.”

Note that because of the introduction of statistical significance, the following modification within the main text was made because of the significance between the no gene control and the (0, 0) state for Figure 2:

“We note that the means of the samples were statistically different in some cases [i.e., no gene to (0,0)], potentially indicating that the distances between the genes are potentially different even when inactive (Figure 2A). Still, overall, these results suggest transcriptional bursting (or a consequence of bursting) is correlated with the formation of promoter-promoter contacts.”

Note, we do not show the modifications to the figure captions here.

(2) If possible, for Figure 1, it would be highly insightful to see whether known enhancer elements are moving closer to promoters of target genes during transcriptions as a comparison to their existing promoter-promoter data.

To investigate whether genes tend to be positioned closer to surrounding enhancer elements with active transcription we mined existing H3K27Ac ChIP-seq data in approximately 200 million cells as described in the following additions to the main text:

“It is conceivable that repositioning is due to enhancer-promoter proximity which might precede transcription activation: the smaller average MPD to the surrounding chromatin with transcription could be due to genes only being active when near surrounding specific enhancers. To investigate we used the density of H3K27Ac as a proxy for enhancer activity. We quantified the density of H3K27Ac ChIP-seq reads within each 50kb segment for IMR90 cells using previously acquired data (Supporting Material) [18]. This analysis resulted in varying densities of H3K27ac throughout Chr21 and is shown in Figure S1A. We then partitioned the H3k27ac density into 4 groups (Low, Med, High, Very High) and investigated the average MPD of each gene to all other loci with and without transcription. Like before (Figure 1), we observed that a gene was indeed closer to the other individual loci when transcriptionally active, but the MPD change did not show a general difference with H3K27ac enrichment when compared to other loci lacking H3K27ac, suggesting that the observed repositioning may not be a result of enhancer-promoter interaction.”

We also added the following analysis specifics to the Supporting Material (Lines:10-19):

H3K27ac Analysis

“To quantify the density of H3K27ac within each corresponding 50kb segment of Chr21 in IMR90 cells, we utilized the ChIP-seq data from the Bing Ren Lab at UCSD:

\\https://www.encodeproject.org/experiments/ENCSR002YRE/. More specifically, we quantified the average number of reads within each 50 kb segment from two biological repeats --- this was done using the software packages Samtools and deepTools. We then normalized the reads by dividing by the sum, allowing us to understand these values in relation to the whole --- this is shown in Figure S1A. To understand whether there is a dependence upon the transcriptioninduced repositioning of the genes based on the H3K27ac signal, we then partitioned each locus into 1 of 4 groups (Low, Med, High, Very High, Figure S1A) and quantified the repositioning based off of the H3K27ac density (Figure S1B, colors).”

And added figure S1 to the Supporting Material.

(3) An extension of the author's findings would be that histone marks associated with transcriptional activity (e.g. H3K27ac) would be enriched in chromatin loci that are in close proximity to the promoter when the gene is on. As a control, chromatin loci containing histone marks associated with gene activity (e.g. H3K27me3) would not move much between the on and off state. In other words, for a locus that is closest in proximity to a promoter, it would be very beneficial to measure the degree of H3K27ac (e.g. a mark of enhancer activity) compared to other surrounding loci of greater physical distance. ChIP-seq datasets for a variety of histone marks are available for the authors to perform this analysis.

We believe that this point is similar to point number 2 above --- as we used the H3K27ac signal as a proxy for enhancer activity. Interestingly, we did not find a difference in the movement between loci with and without transcription as a function of H3K27ac signal (See above).

(4) The changes in MPD stated in Figure 1I seem to be confined to a small region within 50Kb. How would the data look in Figure 1J/K if smaller bin sizes (e.g. 50Kb) were chosen instead of 500Kb?

There may be some confusion with the labeling of the axes in Figure 1D:I. Previously, the axis scaled in 50kb increments - unfortunately, the maximum value within the plots was also around 50, which is confusing to the reader. To make this point clearer we modified the figure.

(5) Given the authors findings on chromosome dynamics obscuring true correlation, it would be helpful to see if other datasets exist that measure the diffusion of a locus when the gene is turned on as comparison to the TFF1 mobility. Can authors compare the diffusion of MS2- labeled intronic sequences where they have a much larger dataset to draw upon?

We have now additionally quantified the diffusion of four other single-allele tagged clones (MYH9, RAB7A, CANX, SLC2A1), as described in the following change to the main text:

“We subsequently performed a similar analysis with the previously published live-cell transcriptional bursting data of four different genes and obtained similar results but with slightly varying diffusion coefficients (Figure S4) [52]. Taking into account the multiple diffusing alleles within the *TFF1* data (Supporting Material), the four diffusion coefficients of the single locus genes range from about.25 x D_TFF1_ up to 1 x D_TFF1_. Lastly, we ultimately decided to proceed with the diffusion coefficient of *TFF1* due to the natural cell movement correction and the relative similarity with the other diffusion coefficients.”

These data are included in the an additional figure S5 within the Supporting Material.

We have also added a brief simulation showing how the diffusion coefficient increases when considering the distance between loci. Note, that this mutual diffusion must be considered due to the fact that the co-bursting within the model is dependent upon the distance between a pair of loci (Supporting Material, Lines:20-39).

Method specifics for single locus diffusion

“To investigate the diffusive behavior of transcriptionally active genes that were tagged at a single allele, we utilized the live-cell microscopy data for four different genes (*MYH9, RAB7A, CANX, SLCA1*) from Wan et al. Of note, this data is different from the multi-allele diffusion analysis within the main text in that there was no internal nuclear reference point to correct for cellular movement over these long timescales. Still, in order to try and correct for the cellular movement we segmented the nucleus using the background GFP signal resulting in a binary image of which pixels belonged to the nucleus and which did not. We then utilized the center of mass of the nucleus of the cell to adjust the diffusive trajectory within that cell.”

Simulation for single and double locus diffusion

“To understand how the diffusion of the single-allele genes relate to the multi-allele *TFF1* data within the main text, we sought to utilize a simple 2d random diffusion model to simulate the diffusive behavior of the two. This is important as the diffusion coefficient we seek to capture for the model is the distance between two different chromosomal loci. To do this we simulated a simple random 2d walks consisting of either 1 particle or 2 particles with 1000 individual trajectories each with a time of 10,000 seconds. Each of the particles was simulated with a diffusion coefficient approximately equal to that of *RAB7A* (D=.1e-3 µm^2^/s). When we quantified the diffusion coefficients of the single particles by fitting the 2d MSDs of the simulated data it resulted in the proper diffusion coefficient (Figure S5B). Then when we quantified the diffusion of the simulations with 2 particles --- taking the distance of one relative to the other, similar to that of *TFF1* --- the MSD resulted in a coefficient approximately double (D=.2e-3 µm^2^/s) --- suggesting that the diffusion of the single-locus data is more similar to the *TFF1*.”

How does this mobility compare with dCas9 measurements examining diffusion of loci that presumably aren't transcribing.

The D apparent for the inactive is about.0023 um^2^/s^.5^ (Gu et al. 2018). All our measured diffusion coefficients are significantly slower than this result (about 1/10^th^). Here we should note that there could be many different reasons for this discrepancy and that the faster the diffusion the more significant the correction to the correlation distance curve will be (Figure 4).

(6) Representation of figures should be improved for increased clarity (e.g. Figures 1J/K, 2, 3A-C, 4 have data cutoff).

We have increased the y-axis ranges of Figure 1J+K and Figure 2A, to avoid the data cutoff problem. However, for the rest of the figures, the y-axis ranges were specifically chosen to illustrate the general trends within the data.

(7) As a way of orienting a non-specialist reader, it might be very helpful to see a representative tracing map of chromatin/promoter loci centroid repositioning upon transcriptional activity.

We added an illustration within the new figure (Figure 5).

(8) One way to increase the general impact of this type of study is to lean into the fact (e.g. further emphasize in the text) that more big imaging datasets are on the way. As such, this study is a good example that re-examining publicly available datasets in a new way can lead to fundamental new insights or answers to long standing questions in the field.

We modified the discussion by adding the following:

“At last, we should also note that more datasets from large-scale microscopy studies are likely on the way, where similar approaches to this study can be taken.”

Reviewer #2 (Recommendations for the authors):I think that the following points, if addressed, can further strengthen a very interesting manuscript.– The analysis is now confined to "on" (1) and "off" gene expression (0) states. I am wondering if the data provide the possibility to stratify the "on" genes in at least "low" and "high" categories and repeat analysis. These categories could reflect high/low FISH signal and/or high/low bursting frequency in the population (something the authors try to incorporate via their live-cell data; see my other comment below).

Unfortunately, we were unable to obtain the data showing high/low FISH signal. But we did perform a number of analyses partitioning the genes into high activity genes and low activity genes. This analysis resulted in the following modifications to the main text:

“We then sought to assess whether the positioning of the genes toward the centroid was dependent upon transcriptional activity. To investigate, we partitioned the available genes into low activity or high activity depending upon whether fractional occupancy was below or above the median, and then performed the above analysis on each subset of genes. That is, the activity of a gene was determined from the fraction of chromosomes where that gene was active. Interestingly, we found that high activity genes were both less variable (Figure S2A) and showed greater movement with active transcription when compared to the low activity genes (Figure S2B and S3). Upon closer inspection (Figure S3A) the greater movement for the high activity genes was not so much due to a different distance to the local chromatin centroid when active but was instead due to larger distances from the centroid when inactive --- this is illustrated by the first genomic distance bin in Figure S3A by comparing the first genomic distance bin of the low activity genes to the high activity genes. In brief, these results suggest that these processes additionally vary depending upon a genes activity level.”

And

“Lastly, we sought to probe the extent to which this phenomenon was dependent upon transcriptional activity (low vs. high as described above). As before, we performed the same analysis but on the two groups of genes separately. Again, the distance change between genes was stronger for more active genes, suggesting these processes also vary depending upon the transcription activity level (Figure S4). Of note for high activity genes, nearly all of them move away from each other when they were separated by large MPD (>1300 nm), suggesting the process of moving to a different location for transcription may be more deterministic for highly active genes (Figure S4E).”

And resulted in figures S2, S3 and S4 within the Supporting Material.

– For the analysis in Figure 3, contact frequency is deduced using high-resolution Hi-C data (not clear to me which and at which resolution to match that of the imaging). However, it is now well understood that Hi-C is generally depleted from promoter-promoter contacts, and that promoter-capture "C" data can prove tricky to quantify and can carry biases. On the other hand, Micro-C data would work very well here and might even reconcile the imaging with "C" technologies.

Note here that we define quantified contact frequency within this work according to the following (new line numbers, but also within original version):

“Here we defined the contact frequency between two genes as the proportion of chromosomes with distances less than 200 nm between the genes' chromosomal segments using the chromatin tracing data.”

– Finally, regarding the (otherwise commendable) effort to generate a model that allows them to "merge" live-cell with fixed-cell data, the authors (i) make a number of assumptions that can be debated, and (ii) use a perhaps too parsimonious way to model chromatin behaviour. As regards (i), a key example is the generalisation of parameters based on analysis of a single locus, TFF1. Similarly the generalisation of ~13 min time for nascent RNA decay probability for all genes based on the MS2 FISH data from ref. 49 is not clear to me. As regards (ii), I think we must acknowledge that in silicon models of chromatin (also linked to transcription output, like the recent Brackley et al., 2021 Nat Commun paper by the Marenduzzo lab) from a number of labs (Mirny, Marenduzzo, Nicodemi, etc.) are growing more and more complex and approximate chromatin and gene expression behaviour evermore accurately. The model employed here is empirically tuned to match aspects of the data, but does not simulate many of the mechanisms known to work on chromatin (like extrusion, which the authors specifically also refer to). I would also like to note that this part of the paper is the least approachable to the average reader, leaves some concepts without any explanation and would benefit from some rewriting; the Results should describe the essence of the model and its key assumptions clearly, and the more complicated math and jargon should be detailed in the Methods, in my view.

At their root, the various models are quite similar in that they introduce a potential on top of random motion to account for various mechanisms, and we believe that our approach to directly incorporate a time-independent potential to capture the diverse empirical data was the most straight forward approach given the available information. The central difference between the different models is how the potentials evolve within the time domain and where (the exact details which are currently lacking) - the most obvious example of a time dependent process being loop extrusion.

To address the specific numbers above:

(i) Our generalization of the ON time comes from live-cell imaging. The justification for using a constant on time within the model mainly comes from the average ON times of various genes being found to range from about 5 to 17 min, with the majority clustered around 12 min (see Wan et al. 2021 Figure 3B (previously ref 49)). To make this assumption clearer to the reader we have added the following to the main text:

“This assumption is motivated by our recent work on high throughput imaging of hundreds of human genes labeled at their endogenous loci using MS2 stem loops --- where it was found the majority of genes had an average on times between 10 to 15 min [52]. Again, we note that this is an assumption due to our lack of temporal information.”

(ii) As stated above, we do agree that future work with more complicated in silico models of this phenomena should be done and that we should acknowledge more specific recent work that could be incorporated into future modifications of our work here.

To do this we have added the following to the main text as well as the new reference:

“Lastly, we should note here that we consider this methodology a first theoretical step, due to the lack of information about the underlying mechanisms on the chromosomal scale. Therefore, future work should be the adaption of more complicated chromatin polymer models to refine our understanding of this phenomena - of special note are those that explicitly model the links between chromatin organizations and their influence on transcription regulation [Brackley et al]. These future models will likely need to explicitly model the underlying processes (like loop extrusion) to capture the variability in chromatin structure and dynamics whose specifics are likely to emerge in future studies - either validating or suggesting modifications to our approach above.”

(iii) To move more of the mathematical “jargon” to the methods section we had to modify the last Results section of the manuscript with the following:

“To account for co-expression, we modeled nascent RNA production as coming either from a co-burst or from an individual burst, where the likelihood that a co-burst or an individual burst occurs is dependent upon the distance between the two genes (Methods). More specifically, the fact that a pair of genes have differing expression levels allowed us to model the proportion of transcription events that are co-bursts with the incorporation of the function w(r_ij_(t)); which is a function of distance between the genes and ranges between 0 and 1. For a pair of genes where the burst frequency of gene i is less than gene j, w(r_ij_(t)) is the proportion of gene i's transcriptional bursts that are co-bursts at each distance (Methods). If the expression levels of the two genes are approximately equal w(r_ij_(t)) is equal to the proportion of bursts that are co-bursts at a given distance for both genes.”

And moved lines 479-497 to the methods.

– Last, I would like to note that the 400 nm cutoff deduced here is not at all unreasonable given previous data on "transcription factory" sizes (diameters between 85-250 nm) and the resolution of the analysed data. Mention of these in the Discussion could strengthen the postulation by the authors. Their statement reading "enrichment in co-bursting for genes separated by < 622 nm suggests the working distance of the underlying mechanism is not direct contact" should be accordingly tuned. Also, a comparison to the sizes of "condensates" would be welcome.

Respectfully, we appreciate the comment but prefer to remain agnostic on the condensate issue.

Nonetheless, I was very happy to see that the manuscript offers a very balanced interpretation of results, previous work, and existing caveats, and was very nice to read overall.

Thanks

Final note: due to outside discussions based on our pre-print, we have added reference 17 (Line 40).